# SketchEvo: Leveraging Drawing Dynamics for Enhanced Image Synthesis

**Zhixin Feng**[1], **Runan Yin**[1], **Lan Yang**[1,2], **Kaiyue Pang**[2] , **Ke Li**[1,2,3,*] **Honggang Zhang**[1], **Yi-Zhe Song**[2]
[1]Beijing University of Posts and Telecommunications, China
[2]SketchX, CVSSP, University of Surrey, United Kingdom
[3]Chenxi Shuzhi (Beijing) Technology Co., Ltd. China

`{fengzhixin, 15735838880, ylan, like1990, zhhg}@bupt.edu.cn`
`kaiyue.pang1993@gmail.com`
`y.song@surrey.ac.uk`

## Abstract

Sketching represents humanity's most intuitive form of visual expression – a universal language that transcends barriers. Although recent diffusion models integrate sketches with text, they often regard the complete sketch merely as a static visual constraint, neglecting the human preference information inherently conveyed during the dynamic sketching process. This oversight leads to images that, despite technical adherence to sketches, fail to align with human aesthetic expectations. Our framework, SketchEvo, harnesses the dynamic evolution of sketches by capturing the progression from initial strokes to completed drawing. Current preference alignment techniques struggle with sketch-guided generation because the dual constraints of text and sketch create insufficiently different latent samples when using noise perturbations alone. SketchEvo addresses this through two complementary innovations: first, by leveraging sketches at different completion stages to create meaningfully divergent samples for effective aesthetic learning during training; second, through a sequence-guided rollback mechanism that applies these learned preferences during inference by balancing textual semantics with structural guidance. Extensive experiments demonstrate that these complementary approaches enable SketchEvo to deliver improved aesthetic quality while maintaining sketch fidelity, successfully generalizing to incomplete and abstract sketches throughout the drawing process.

## 1 Introduction

Sketching is one of humanity's most intuitive forms of visual communication. The act of drawing – translating mental concepts into physical strokes – naturally encodes rich information not just in the final result, but in the sequential process itself. Recent advances in diffusion models (Ho et al., 2020; Song et al., 2021a; Hu et al., 2024; Song et al., 2021b) have enabled remarkable progress in controllable image generation, with approaches like ControlNet (Zhang et al., 2023), T2I-Adapter (Mou et al., 2024), and VersaGen (Chen et al., 2025) successfully integrating sketch conditions with textual prompts.

However, these methods (He et al., 2024; Liu et al., 2024; Li et al., 2024; Qin et al., 2023b; Hu et al., 2023) primarily focus on the final sketch as a static spatial constraint, overlooking the intermediate information of the drawing process. When processing amateur sketches, existing approaches often produce technically correct but aesthetically disappointing results – they satisfy structural constraints (often poor as per amateur sketches) yet fail to capture human intent. This disconnect reveals a deeper problem: existing models don't understand how humans conceptualize and refine visual ideas through the progressive accumulation of strokes.

Our investigation reveals that the core challenge lies in a fundamental misalignment between generated outputs and human aesthetic preferences. While recent preference alignment approaches like

---

*Correspondence to: like1990@bupt.edu.cn. Code to be found at GitHub page.

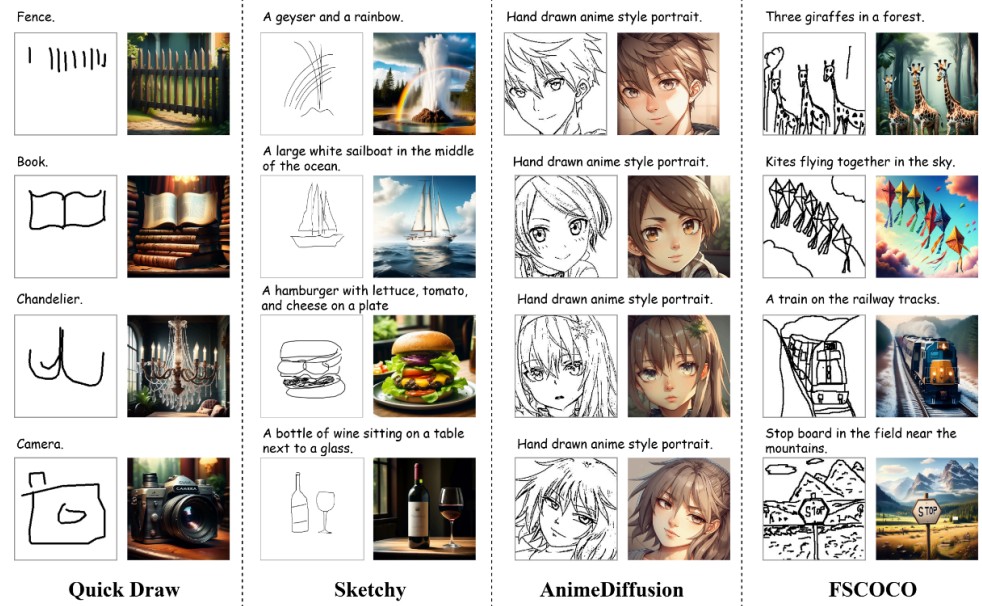

Figure 1: Visualization of SketchEvo results. The model produces images aligned with human preferences, trained only on the Sketchy dataset.

DPOK (Fan et al., 2023), DPO (Rafailov et al., 2023), D3PO (Yang et al., 2024), SPO (Liang et al., 2025), and LPO (Zhang et al., 2025) have improved image quality in other contexts, they face a critical obstacle in sketch-guided generation. These methods optimize models by comparing generated variations and adjusting to produce more preferred outputs, but in multimodal sketch-and-text generation, the dual constraints create insufficiently different samples when using conventional noise-based variations. These limited variations provide weak training signals that struggle to capture meaningful aesthetic improvements while preserving sketch details. This creates a false dichotomy where models must choose between faithfully reproducing potentially flawed amateur sketches or generating visually pleasing but sketch-inconsistent images.

To address this challenge, we introduce SketchEvo, a framework that leverages the sketch sequences – from initial strokes to completion – as a powerful source of diversity for preference-based optimization. Our key insight is that intermediate sketches from different drawing stages represent varying levels of abstraction and detail, offering meaningful semantic and structural divergence while maintaining connection to the user's intent. By using these sketch variations instead of relying solely on noise perturbations, SketchEvo creates meaningfully different sample pairs that provide strong signals for human preference alignment even under tight multimodal constraints.

SketchEvo introduces two complementary innovations that work together across the model lifecycle: First, a sequence-guided sampling strategy transforms sketching information into effective learning signals during training by incorporating sketches at different completion stages as conditional inputs. This expands candidate diversity and creates sample pairs with greater aesthetic divergence, providing more informative gradients for preference optimization. Second, building upon this improved preference alignment, our sequence-guided rollback mechanism applies these learned preferences during inference by leveraging initial sketch strokes to guide rollback. This quantifies information gain from both textual and sketch conditions, ensuring the aesthetic improvements learned during training are fully realized in the generated images while maintaining structural fidelity to the user's sketch intent.

Our extensive experiments demonstrate substantial improvements across multiple human preference metrics and conditional fidelity measures. Particularly noteworthy is SketchEvo's strong generalization ability in on-the-fly sketch-to-image tests, where it successfully produces high-quality images

from incomplete and abstract sketches at various stages of the drawing process, enabling truly interactive sketch-based creation.

## 2 RELATED WORK

### 2.1 CONTROLLABLE IMAGE GENERATION

The rapid development of diffusion models has significantly advanced text-to-image generation (Podell et al., 2024; Esser et al., 2024; Labs, 2024). Building on pretrained T2I models, various conditional strategies (Ye et al., 2023; Ruiz et al., 2023; Chen et al., 2024; Wang et al., 2024) incorporate external structural cues, with sketches serving as an intuitive prior. Early approaches such as ControlNet (Zhang et al., 2023) and T2I-Adapter (Mou et al., 2024) introduce sketch features via auxiliary branches, and ControlNet++ (Li et al., 2024) further refines spatial consistency. Subsequent extensions—including VersaGen (Chen et al., 2025), SmartControl (Liu et al., 2024), and Uni-Control (Qin et al., 2023a)—enhance robustness by expanding controllable modalities or improving feature injection. AnimateDiff (Guo et al., 2024) extends such conditioning into temporal domains, highlighting the utility of structured priors beyond single images. (Koley et al., 2024) improves robustness when sketches are rough, imprecise, or drawn by amateurs, while KnobGen (Navard et al., 2024) tackles sparse and structurally incomplete sketches through a dual-path design that jointly models coarse and fine structural cues. However, most existing approaches treat sketches merely as static inputs, overlooking the rich intermediate information and implicit human preference signals embedded in the drawing process.

### 2.2 HUMAN PREFERENCE OPTIMIZATION

Various aesthetic metrics have been proposed, such as ImageReward (Xu et al., 2023), HPSv2 (Wu et al., 2023), PickScore (Kirstain et al., 2023), and VisionReward (Xu et al., 2024). Recent approaches improve subjective alignment through rollback mechanisms (LiChen et al., 2025), RLHF-inspired formulations (Ouyang et al., 2022; Zhang et al., 2024; Fan et al., 2023; Black et al., 2024; Rafailov et al., 2023; Yang et al., 2024), and preference optimization with sample pairs (Liang et al., 2025; Zhang et al., 2025; Yeh et al., 2024). However, their reliance on random Gaussian noise perturbations limits the capture of structural and layout-level aesthetic intentions.

## 3 PROBLEM STATEMENT AND BACKGROUND

**Problem formulation.** This paper investigates the aesthetic degradation problem in sketch-guided image generation, with inputs being textual descriptions and a entire drawing sequence $\{s_1, s_2, \ldots, s_N\}$ from the first stroke $s_1$ to the completed sketch $s_N$. The goal is to generate an image $I$ that combines high aesthetic quality and sketch fidelity. However, current human preference alignment methods Fan et al. (2023); Wallace et al. (2024); Yang et al. (2024); Liang et al. (2025); Rafailov et al. (2023) face optimization bottlenecks in multimodal generation tasks with dual constraints of text and sketch. The core issue lies in their reliance on single Gaussian noise perturbation, leading to insufficient diversity in generated sample pairs – only subtle differences can be produced under dual constraints, making effective optimization difficult.

**Preference Optimization for Diffusion Models**. Aligning diffusion models with human aesthetic preference is very challenging. Given a winning image $x^w$, a losing image $x^l$, and a text condition $c$, methods such as DPORafailov et al. (2023) and D3POYang et al. (2024) generate intermediate latent image pairs $(x_t^w, x_t^l)$, thereby incentivizing the diffusion model $p_\theta$ to prioritize the generation of $x_t^w$ over $x_t^l$. However, the preference ordering $(x_t^w, x_t^l)$ during the denoising process does not always align with the initial preferences $(x^w, x^l)$. To address this issue, SPO Liang et al. (2025) tackles it by sampling from $x_{t+1}$ to generate a denoised sample candidate pool $\{x_t^1, x_t^2, \ldots, x_t^K\}$, and introduces a Step-Aware Preference Model (SPM) to predict preference scores for samples in the candidate pool, thereby constructing refined preference pairs $(x_t^w, x_t^l)$. The optimization objective of SPO is:

$$L_{\text{SPO}} = -\mathbb{E}_{x_t^w, x_t^l \sim p_\theta(x_t|x_{t+1}, c, t+1)} \left[\log \sigma \left(\beta \triangle r\right)\right] \tag{1}$$

$$\triangle r = \frac{p_\theta(x_t^w \mid x_{t+1}^w, c, t+1)}{p_{\text{ref}}(x_t^w \mid x_{t+1}^w, c, t+1)} \Big/ \frac{p_\theta(x_t^l \mid x_{t+1}^l, c, t+1)}{p_{\text{ref}}(x_t^l \mid x_{t+1}^l, c, t+1)} \tag{2}$$

where the symbol $\sigma$ denotes the sigmoid function, $\beta$ is a regularization hyperparameter and $p_{\text{ref}}$ denotes the reference value from the fixed initial denoising model $p_\theta$.

**Construction of Denoised Sample Candidate Pool**. SPO constructs the denoised sample candidate pool $\{x_t^1, x_t^2, \ldots, x_t^K\}$ by adding random Gaussian noise $z$ on the generated samples $\mu_\theta(x_{t+1}, c, t+1)$, as formalized below:

$$x_t^k = \mu_\theta(x_{t+1}, c, t+1) + z \tag{3}$$

where $z \sim \mathcal{N}(0, I)$, $x_t^k$ denotes the $k$-th sample in the candidate pool. Sample diversity of candidate pool is solely governed by the noise $z$ sampled from a standard Gaussian distribution.

## 4 PROPOSED METHOD

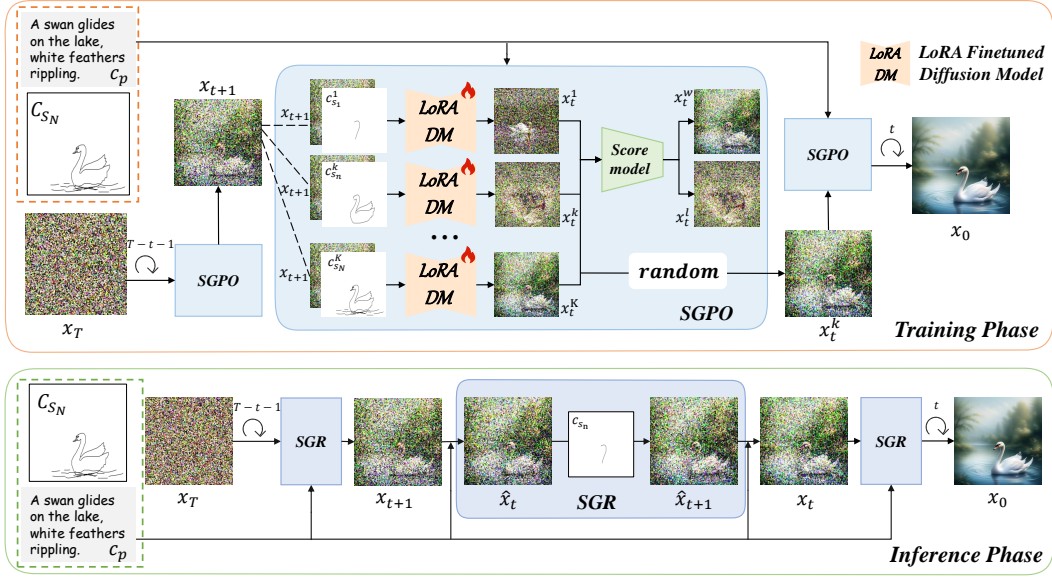

Figure 2: The framework of SketchEvo. During training, the SGPO module can obtain distinctive positive-negative sample pairs, fine-tuning the LoRA model in the U-Net to align with human preferences. During inference, a SGR mechanism is employed to strengthen conditional information.

### 4.1 SEQUENCE-GUIDED PREFERENCE OPTIMIZATION (SGPO)

**Sequence-Guided Sampling Strategy.**

Unlike traditional methods that solely rely on random noise perturbations to generate candidate samples, we introduce sketch sequences $\{s_1, s_2, \ldots, s_N\}$ from the drawing process to construct a dynamically evolving candidate pool. Specifically, as shown in Fig. 2, at each sampling step, $K$ sketches are randomly selected from the sketch sequence as conditions to guide the model in constructing a candidate pool $\{x_t^1, x_t^2, \ldots, x_t^K\}$ through denoising $x_{t+1}$. This process is formally expressed as:

$$x_t^k = \mu_\theta(x_{t+1}, c, c_{s_n}^k, t+1) + z \tag{4}$$

where $c_{s_n}^k$ denotes the sketch condition $s_n$ adopted for the $k$-th sample $x_t^k$.

Within the dynamically evolving candidate pool, differences between samples are no longer entirely dominated by random noise $z$. We leverage sketches at different completion stages as conditional inputs to guide the denoising process, thereby generating diversified denoised samples. Since sketches at each intermediate stage undergo significant evolution in structural and detail levels (As shown in Fig. 2, within the swan sketch drawing sequence, substantial discrepancies exist between $s_1, s_n, s_N$), this approach enables the generation of diversified samples.

**Positive-Negative Sample Pair Selection Strategy.** At each diffusion timestep $t$, we utilize a pretrained scoring model to evaluate the samples in the dynamic evolving candidate pool. The highest- and lowest-scoring samples are selected and denoted as $p_\theta(x_t^w \mid x_{t+1}^w, c, c_{s_w}, t+1)$ and

$p_\theta(x_t^l \mid x_{t+1}^l, c, c_{s_l}, t+1)$, respectively, forming a positive-negative sample pair. To better optimize the model, only sample pairs surpassing a predefined threshold are utilized for model training.

**The Impact of Candidate Pool Sample Diversity on Preference Optimization.** According to equation 1, given two samples $x_t^w$ and $x_t^l$, the gradient of the reward function with respect to the SGPO, Eq. 5 is reformulated as follows:

$$\nabla_\theta \mathcal{L}_{Ours} = -\beta \mathbb{E}\left[\sigma(-\beta \triangle r)\left(\nabla_\theta \log p_\theta(x_t^w | c_{s_w}, x_{t+1}^w, c, t+1) - \nabla_\theta \log p_\theta(x_t^l | c_{s_l}, x_{t+1}^w, c, t+1)\right)\right]$$
(5)

$$\triangle r = \frac{p_\theta(x_t^w \mid x_{t+1}^w, c, c_{s_w}, t+1)}{p_{\text{ref}}(x_t^w \mid x_{t+1}^w, c, c_{s_w}, t+1)} \Big/ \frac{p_\theta(x_t^l \mid x_{t+1}^l, c, c_{s_l}, t+1)}{p_{\text{ref}}(x_t^l \mid x_{t+1}^l, c, c_{s_l}, t+1)}$$
(6)

where $c_{s_w}, c_{s_l}$ represent the sketches condition corresponding to the best latent and the worst latent, respectively.

As derived from Eq. 5, the effectiveness of gradients depends on the discrepancy between $x_t^w$ and $x_t^l$. When sample diversity is insufficient, this discrepancy diminishes, causing $\triangle r \to 1$, which degenerates gradients into an uninformative signal with poor directionality. It can be observed that our proposed SGPO significantly outperforms SPO in candidate pool sample diversity. This disparity is directly reflected in the differences between sample pairs — SGPO generates positive-negative sample pairs with more pronounced disparities. The higher sample diversity enables SGPO to provide richer gradient information for the model during the optimization process.

## 4.2 Sequence-Guided RollBack (SGR) Mechanism During Inference Phase

To effectively transfer the improvements of SGPO from the training phase to the inference phase, we introduce a sequence-guided rollback mechanism. Previous works have demonstrated that incorporating a rollback mechanism into the text-guided image generation process allows the latent encoding to more effectively capture semantic information. The cumulative latent difference in rollback can be formulated as: where $h_t = \sqrt{1/\alpha_t - 1} - \sqrt{1/\alpha_{t-1} - 1}$, $u_\theta$ is the model-predicted noise, $\tau_1(t)$ denotes the semantic information gain term and $\tau_2(t)$ denotes the error term, which can be neglected under certain conditions.

When directly transferring existing rollback mechanism to sketch-to-image generation tasks, their performance improvement faces significant bottlenecks. The fundamental reason lies in the fact that methods design optimization objectives solely for uni-modal textual conditions, failing to model the structural priors inherent in sketch conditions. To address this issue, we propose the SGR mechanism, which is implemented by integrating the sketch-drawing sequence and text conditions to jointly guide the rollback process, as formulated below:

$$\epsilon_\theta^t(x_t) = (1 + \gamma_1)u_\theta(x_t, c, c_{s_N}, t) - \gamma_1 u_\theta(x_t, \emptyset, c_{s_N}, t)$$
$$\epsilon_\theta^t(\tilde{x}_t) = (1 + \gamma_2)u_\theta(\tilde{x}_t, c, c_{s_n}, t) - \gamma_2 u_\theta(\tilde{x}_t, \emptyset, c_{s_n}, t)$$
(7)

In our method, the information gain encompasses not only semantic information from text conditions $c$, but also structural and detailed information introduced by the sketch sequences $s_n$, as well as human preference information learned by the model parameters $\theta$. During the inference phase, when the sketch sequence $s_n$ is predetermined, our rollback mechanism is simplified to follow the generation manner of text-to-image. Consequently, we adopt the configuration of $\gamma_1$ and $\gamma_2$ to maximize semantic information enhancement. With $\gamma_1$ and $\gamma_2$ fixed, the information gain is determined by the following formula (detailed proofs are provided in Appendix D):

$$\delta_{Z-Sampling} \propto \sum_{t=1}^T (\tau_1(t))^2 \propto \sum_{t=1}^T (u_\theta(x_t, c, c_{s_n}, t))^2$$
(8)

The discrepancy in structural and detailed information gain is determined by the noise generated through the control of textual conditions $c$ and sketch conditions $c_{s_n}$ in the generative model. By increasing the divergence between $c$ and $c_{s_n}$, the cumulative information gain can be augmented.

The SGR mechanism quantifies the information gain from text and sketch conditions, ensuring that aesthetic enhancements learned during training are fully manifested in the generated images while preserving structural fidelity to the user's original sketch intent.

Table 1: Quantitative results. We evaluate our method and competitors on the sketchy dataset in terms of human preference alignment and conditional fidelity.

| Method | Human Preference Alignment | | | Conditional Fidelity | |
|---|---|---|---|---|---|
| | Image Reward ↑ | HPS v2 ↑ | Pick score ↑ | LPIPS-sketch ↓ | CLIP-Score ↑ |
| ControlNet (Zhang et al., 2023) | 0.004 | 24.08 | 20.03 | **0.11** | 23.70 |
| T2i-Adapter (Mou et al., 2024) | -0.001 | 23.60 | 20.58 | 0.20 | 15.86 |
| VersaGen (Chen et al., 2025) | 0.08 | 24.68 | 20.79 | 0.14 | 23.77 |
| ControlNet++ (Li et al., 2024) | -0.011 | 24.19 | 20.56 | 0.13 | 23.83 |
| AnimateDiff (Guo et al., 2024) | 0.23 | 23.68 | 20.42 | 0.14 | 23.56 |
| UniControl (Qin et al., 2023a) | 0.024 | 23.08 | 20.39 | 0.13 | 23.75 |
| ControlNet-DPO (Rafailov et al., 2023) | 0.47 | 25.02 | 20.86 | 0.16 | 23.35 |
| ControlNet-SPO(Liang et al., 2025) | 0.61 | 27.69 | 22.04 | 0.17 | 23.65 |
| SGPO | 1.03 | 28.87 | 21.94 | 0.20 | 23.86 |
| Ours | **1.18** | **30.08** | **22.41** | 0.15 | **24.15** |

# 5 EXPERIMENT

## 5.1 EXPERIMENTAL SETUPS

Our method is fine-tuned based on the ControlNetXL (Podell et al., 2024) to achieve alignment with human preferences. All experiments are conducted on NVIDIA A100 GPUs for both training only with Sketchy (Sangkloy et al., 2016) dataset . Detailed information about data processing, hyperparameters employed in the experiments are provided in the Appendix A and B.

## 5.2 COMPARISON WITH THE STATE-OF-THE-ART METHODS

To verify the effectiveness of our method, we compared it with state-of-the-art multi-condition controlled image generation models ControlNet (Zhang et al., 2023), T2i-Adapter (Mou et al., 2024), VersaGen (Chen et al., 2025), ControlNet++ (Li et al., 2024), AnimateDiff (Guo et al., 2024), Unicontrol Qin et al. (2023a) and preference aligning methods DPO (Rafailov et al., 2023) and SPO Liang et al. (2025) under their original settings. Since the original DPO and SPO are based on SDXL (Podell et al., 2024) fine-tuning and only accept text conditions, we replaced the SDXL model in ControlNet with their fine-tuned SDXL models. SGPO refers to our proposed method without employing the rollback mechanism.

For Human Preference Alignment, as shown in Tab. 1, We found that, compared with multi-condition controlled image generation models, methods after preference alignment achieved aesthetic score improvements, with ours outperforming ControlNet-DPO and ControlNet-SPO. From a detailed view, human preference is reflected in three aspects: i) **Position**: As shown in the first row of Fig. 3, The cat and chair generated by our model exhibit a more natural positional relationship, demonstrating superior spatial layout preference capability. ii) **Composition**: As presented in the second row of Fig. 3, the cake generated by our model exhibit proportionally more realistic scaling relative to the candles, demonstrating superior compositional preference capability. iii) **Color**: As illustrated in the third row of Fig. 3, our model accurately identifies the color boundaries between foreground and background, altering only the helicopter's main body to a camouflage pattern while preserving the background's original appearance, demonstrating precise control over color preferences.

Table 2: Quantization results demonstrating the model's generalization capability when trained only on the Sketchy dataset and evaluated on unseen data.

| Dataset | Method | Image Reward ↑ | HPS v2 ↑ | PickScore ↑ | LPIPS -Sketch ↓ | CLIP Score ↑ |
|---|---|---|---|---|---|---|
| QuickDraw | ControlNet | -0.56 | 15.24 | 17.17 | **0.47** | 23.07 |
| | ControlNet-SPO | 0.40 | 27.05 | 21.38 | 0.67 | 24.01 |
| | Ours | **0.86** | **30.22** | **21.67** | 0.68 | **24.28** |
| AnimeDiffusion | ControlNet | -0.11 | 18.34 | 19.62 | **0.10** | 23.17 |
| | ControlNet-SPO | 0.27 | 23.28 | 19.67 | 0.15 | 23.99 |
| | Ours | **1.32** | **31.57** | **23.68** | **0.10** | **24.96** |
| FSCOCO | ControlNet | -0.03 | 23.13 | 18.99 | **0.41** | 24.05 |
| | ControlNet-SPO | 0.52 | 27.51 | 19.09 | 0.50 | 24.39 |
| | Ours | **0.96** | **30.31** | **21.78** | 0.45 | **24.77** |

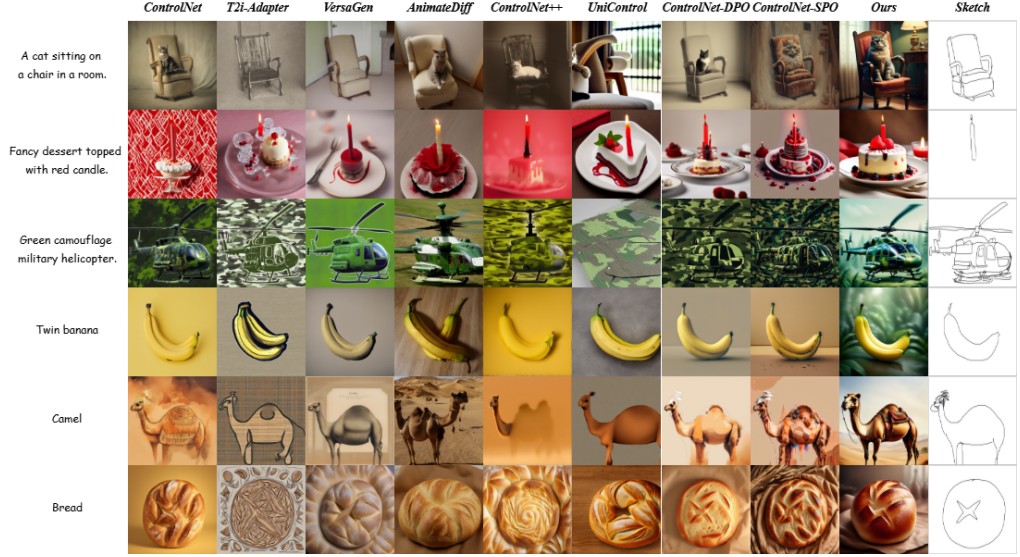

Figure 3: Visualized comparison of ControlNet, T2I-Adapter, VersaGen, AnimateDiff, Control-Net++, UniControl, ControlNet-DPO, ControlNet-SPO, SGPO and Ours on Sketchy dataset.

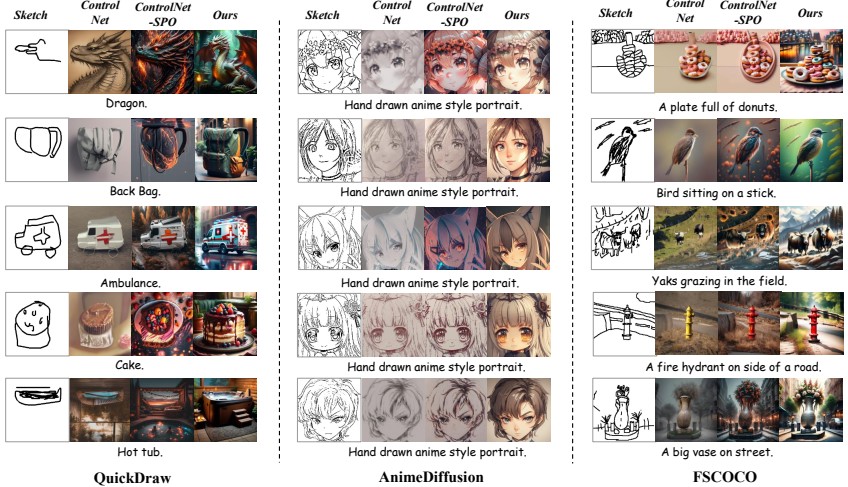

Figure 4: Generalization results are visualized, consistently generating high-quality images regardless of sketch expertise. Images from more professional sketches show even better alignment with inputs.

As shown in Tab. 1, our method achieves the highest semantic fidelity, though not the highest sketch similarity, as ControlNet and T2I-Adapter strictly follow sketches. Compared to preference-alignment methods like DPO and SPO, our sequence-guided approach better integrates text and sketch information, preserving fine-grained structures while generating natural images. As illustrated in Fig. 3 (rows 4-6), it simultaneously: (i) accurately realizes quantitative semantics (e.g., two bananas); (ii) suppresses redundant strokes (e.g., camel head) via enhanced semantic supervision; (iii) enhances visual expressiveness while maintaining precise text-image consistency.

## 5.3 GENERALIZATION ABILITY OF THE MODEL

To further evaluate the generalization ability of our model, we conduct inference on the Quick-Draw (qui, 2016), AnimeDiffusion (Cao et al., 2024), and FSCOCO (Chowdhury et al., 2022) datasets. As shown in Tab. 2 and Fig. 4, our model consistently achieves competitive performance

across diverse datasets, demonstrating robust generalization beyond the training domain. More results demonstrations are presented in G.

For single-object sketch datasets, the model adaptively balances aesthetic quality and sketch similarity according to the level of sketch abstraction, without requiring manual weight adjustment. As shown in Tab. 2, whether tested on the abstract QuickDraw dataset or the professional AnimeDiffusion dataset, our model consistently produces results with the highest aesthetic scores while maintaining strong sketch fidelity. For example, on the QuickDraw dataset, ControlNet generates flipped hot tub images that defy real-world structure, while ControlNet-SPO yields higher aesthetic scores but poor sketch alignment. In contrast, our model automatically integrates both sketch consistency and aesthetic quality, generating superior images without manual tuning.

For complex scene sketch datasets, our model consistently achieves higher aesthetic scores and sketch similarity. As shown in Tab. 2, on FSCOCO, our method outperforms ControlNet-SPO in both metrics. Due to the presence of multiple elements in scene sketches, overall similarity scores are lower, even ControlNet achieves only 0.41, underscoring the difficulty of strict alignment in complex scenarios.

Moreover, higher sketch professionalism leads to greater image-sketch similarity. As illustrated in Fig. 1 and Fig. 4, the similarity between images and sketches progressively improves from the less professional QuickDraw, through sketch and FSCOCO, to the highly professional Animeduffusion dataset. Notably, this enhancement in similarity is achieved without compromising aesthetic quality, demonstrating strong preservation of sketch details.

## 5.4 ABLATION STUDY AND ANALYSIS

**Can Sequence-Guided Sampling Strategy enhance the sample diversity of the candidate pool?**
Indeed, our method employs a higher positive-negative sample filtering threshold compared to the ControlNet-SPO (0.8 *v.s.* 0.4). To visually validate the enhanced diversity of the candidate pool, Fig. 5(b) specifically compares the maximum aesthetic score differences in the candidate pools between the two methods across different denoising stages. Experimental results demonstrate that our method achieves significantly higher maximum aesthetic score differences than ControlNet-SPO at all denoising stages, directly confirming that the sequence-guided sampling strategy effectively enhances diversity characteristics in candidate samples.

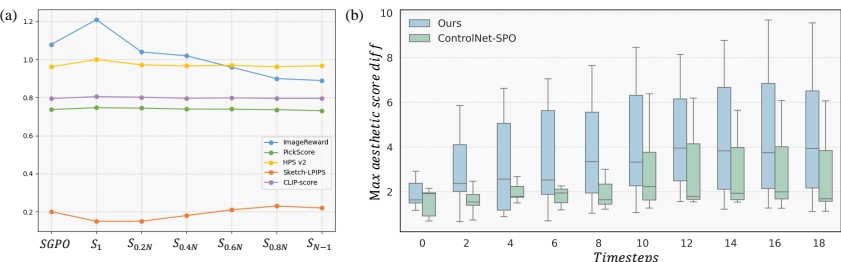

Figure 5: The ablation study results, (a) presents the performance comparison of rollback mechanisms guided by sketches of varying abstraction levels under different evaluation metrics. (b) shows the statistical analysis of score differences between positive and negative samples in the candidate pool.

**The Impact of Different Abstraction Levels of Sketch in SGR Mechanism.**

In Fig. 5(a), we compare the performance of rollback mechanisms guided by sketches at different abstraction levels through multi-dimensional evaluation. For visualization convenience, the values of HPS v2 (Wu et al., 2023), PickScore (Kirstain et al., 2023), and CLIP-Score (Hessel et al., 2021) are scaled down by a factor of 30. Six comparative schemes are designed: SGPO denotes our method without the rollback mechanism; $s_1$ represents the rollback strategy guided by the first sketch stroke; $s_{0.2N}, s_{0.4N}, s_{0.6N}, s_{0.8N}$ correspond to rollback strategies guided by sketches at 20%-80% completion stages; $s_{N-1}$ denotes the strategy guided by the second-to-last sketch stroke. Experimental results reveal significant performance differences across schemes: the $s_{N-1}$-guided strategy performs worst, while the $s_1$-guided mechanism surpasses others in comprehensive evaluation. This phenomenon aligns with the theoretical derivation in Eq. 8 – as the discrepancy between the text condition $c$ and sketch condition $c_{s_n}$ increases, the cumulative information gain can be aug-

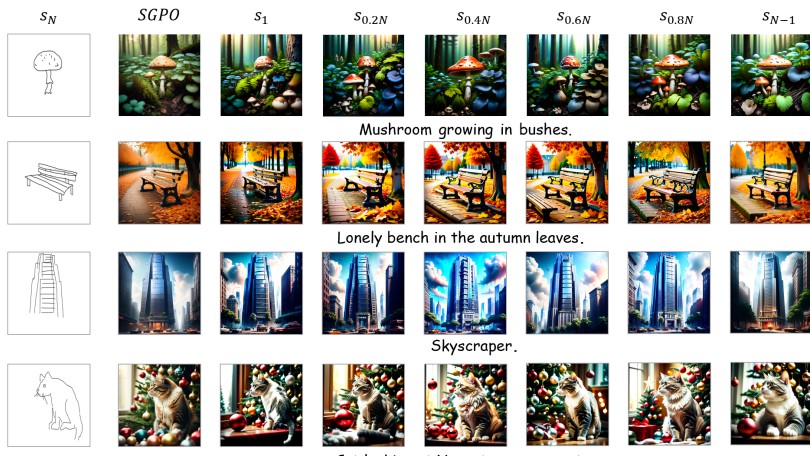

Figure 6: Visualization esults to the impact of different abstraction levels of sketch in SGR mechanism.

mented, thereby effectively amplifying both the semantic-structural information of conditions and the preference information of the model.Visual demonstrations are presented in Fig. 6.

As shown in the Fig. 6, a closer examination reveals that the level of sketch abstraction used for rollback guidance has an impact on the control over image details. When guided by the more abstract sketch $s_1$, the generated results maintain stable quality while better preserving the structural features of the sketch. However, as the number of strokes increases, this consistency gradually declines.

### 5.5 ON-THE-FLY SKETCH-TO-IMAGE GENERATION TEST

During the dynamic process of sketch drawing, an advanced sketch-to-image generation model must demonstrate the capability to produce high-quality images from sketches at any drawing stage. To evaluate this, we propose the On-the-fly Sketch-to-Image Generation Test . Leveraging MasaCtrl's (Cao et al., 2023) foreground editing strategy, we effectively stabilize the generated background to eliminate its interference. As shown in Fig. 7, our method generalizes well: even with highly abstract sketches or only a few strokes, the generated images align with human preferences and accurately reflect input conditions. As sketches gain detail, the outputs evolve accordingly, demonstrating the model's practical value in interactive creation.

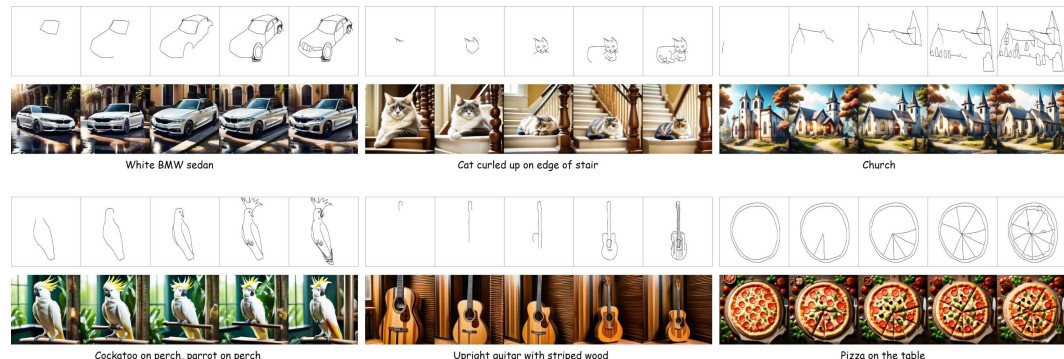

Figure 7: Our method generates natural, conditionally accurate images from sketches of varying abstraction, with results increasingly matching user expectations as drawing progresses.

### 6 CONCLUSION

In this paper, we propose a novel framework named SketchEvo, which leverages drawing dynamics to enhance image generation. Specifically, during the training phase, we introduce a sequence-guided optimization strategy to enhance the diversity of candidate samples, thereby improving the alignment of generated results with human preference. In the inference phase, a sequence-guided

rollback mechanism is adopted: the initial sketch strokes are used to guide rollback, ensuring that the generation process aligns with human preferences while meeting user conditions. Experimental results show that our method effectively breaks through the trade-off bottleneck between creativity and structural fidelity in traditional models.

## ACKNOWLEDGMENTS

This work was supported in part by the Postdoctoral Fellowship Program of the China Postdoctoral Science Foundation (CPSF) under Grant No. GZC20251088.

## ETHICS STATEMENT

This research strictly adheres to the ICLR Code of Ethics. All used datasets are publicly available and comply with the relevant licenses and privacy policies. The study does not involve human subjects or sensitive information, and contains no discrimination, bias, or harmful impact. There are no conflicts of interest.

## REPRODUCIBILITY STATEMENT

We place great importance on the reproducibility of this research. Detailed descriptions of the model architecture and key experimental settings are provided in both the main text and the appendix. Anonymized and downloadable source code has been submitted as supplementary material, and can be accessed at: https://anonymous.4open.science/r/SketchEvo6-B34C123/. The preprocessing steps for the datasets used in the experiments, as well as the complete experimental procedures, are thoroughly presented in the supplementary materials. Theoretical assumptions and detailed proofs of the main conclusions are included in the appendix.

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

# Technical Appendices and Supplementary Material

## A   EXPERIMENTAL DETAILS

In this section, we provide additional details on the dataset, experimental setup, and evaluation metrics.

**Datasets.**

Sketchy  (Sangkloy et al., 2016) dataset. The Sketchy dataset comprises 125 object categories, with each category containing 100 real images and corresponding 7-10 hand-drawn sketches along with their drawing sequences, totaling 12,500 real images and 75471 associated sketches. To ensure experimental rigor, we employ a stratified sampling strategy to partition the data into non-overlapping random splits at an 8:2 ratio.

FSCOCO  (Chowdhury et al., 2022) dataset contains 10,000 scene sketches drawn by 100 non-professionals, each with stroke-level temporal data and text descriptions. Collected under a memory-based drawing setup, it captures both complex structures and human subjective preferences, making it well-suited for evaluating models in multi-object scene generation.

QuickDraw  (qui, 2016) is a large-scale sketch dataset from Google's "Quick, Draw!" game, containing 50 million sketches across 345 categories drawn by millions of non-expert users. Constrained to 20 seconds, the sketches are highly abstract, often with distortions or missing parts, creating a clear gap from real images. In this work, we use a 2.5K subset to evaluate our model's ability to generate images under such abstract conditions, focusing on its alignment with human preferences and its capacity to interpret and repair incomplete structures.

AnimeDiffusion (Cao et al., 2024) dataset is a benchmark specifically designed for anime face line-art colorization, created through face cropping, alignment, and denoising. It consists of 31,696 training samples and 579 test samples, with a resolution of 256×256. Each colored face image is paired with a corresponding line drawing generated using the XDoG algorithm, featuring diverse line styles and a high level of professionalism, making it well-suited for evaluating model performance under professional sketch conditions.

**Settings.**

During training, we employ the Adam optimizer with a learning rate of $1 \times 10^{-5}$, a batch size of 4, and perform 4 training epochs. The diffusion model is configured with a total of 20 timesteps, and the candidate pool size $K$ is set to 5. The initial sketch $s_1$ and the final complete sketch $s_N$ are always selected, while 3 additional sketches are chosen from the remaining sequence as conditions to guide the denoising process for generating candidate pool samples. A threshold of 0.8 is applied for filtering positive-negative sample pairs, with timesteps 10 to 20 designated as the late denoising phase.

**Metrics.**

PickScore  (Kirstain et al., 2023) developed Pick-a-Pic, a large open dataset consisting of textto-image prompts and real user preferences for generated images. They then utilized this dataset to train a CLIP-based scoring function, PickScore, for the task of predicting human preferences.

ImageReward  (Xu et al., 2023) developed ImageReward, the first general-purpose text-to-image human preference reward model, which is trained based on systematic annotation pipeline, including rating and ranking and has collected 137,000 expert comparisons to date.

HPS v2  (Wu et al., 2023) first introduced the Human Preference Dataset v2 (HPD v2), a largescale dataset comprising 798,090 human preference choices on 433,760 pairs of images. By finetuning CLIP using HPD v2, they developed the Human Preference Score v2 (HPS v2), a scoring model that more accurately predicts human preferences for generated images.

Sketch-LPIPS  (Zhang et al., 2018) is used to evaluate the perceptual similarity between a generated image and a sketch. It extracts the target region from the generated image using SAM  (Kirillov et al., 2023), computes its Canny edge map, and compares it with the original sketch. Then, features

are extracted using a deep network (e.g., VGG) to calculate the LPIPS score, where a lower score indicates higher similarity.

CLIP-Score (Hessel et al., 2021) is a metric used to measure the semantic consistency between an image and its text description. It uses the CLIP model to extract features from both the image and the text, then computes the cosine similarity between them. A higher score indicates better alignment between the image and the text.

## B    ABLATION

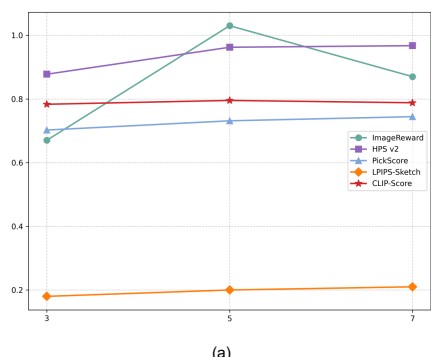
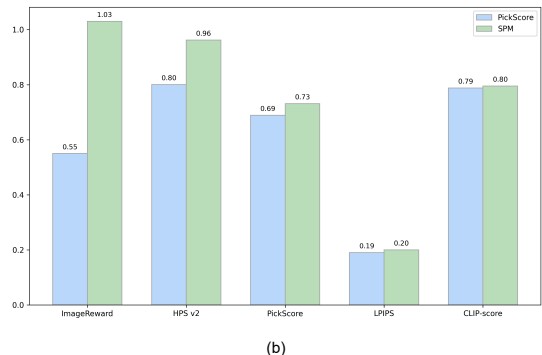

(a)                                                          (b)

Figure 8: Ablation study results: (a) illustrates the effect of candidate pool size on generation quality under the SGPO mechanism, while (b) shows the impact of different scoring models on the experimental outcomes.

**Ablations on how SPO performance change w.r.t. sample diversity.**

As we can see Fig. 8 (a), when $k$ increases from 5 to 7, the marginal performance gain is substantially lower than that observed between $k = 3$ and $k = 5$, revealing an inherent optimization bottleneck. The underlying mechanism is attributed to the progressive homogenization of sketch states: as more intermediate sketches are sampled, the state differential between $s_n$ and $s_{n-i}$ decays, driving performance improvements toward an asymptotic plateau. Increasing $k$ also incurs a significantly higher computational cost for candidate pool construction, reducing training efficiency. In summary, considering both sampling efficiency and generation quality, we adopt $k = 5$ as the default setting.

**The impact of scoring model.**

To evaluate the sensitivity of our method to the choice of scoring model, we conducted an additional experiment using PickScore as the scoring model. The results of this experiment are presented in the Fig. 8 (b). Our method shows strong robustness to the choice of scoring model, as evidenced by the small performance differences across most metrics (HPS v2, PickScore, LPIPS, and CLIP-Score) when evaluated with different scoring models.

## C    PSEUDOCODE FOR THE INFERENCE STAGE IN THE SGR MECHANISM

---

**Algorithm 1** Inference Stage of SGR Mechanism

---

**Require:** Text prompt $c$, Complete sketch $c_{s_N}$, Guidance sketch $c_{s_n}$, Inference steps $T$
**Ensure:** Clean image $x_0$
 1: $x_T \sim \mathcal{N}(0, I)$
 2: **for** $t = T$ **down to** $1$ **do**
 3:     $x_{t-1} \leftarrow \Phi^t(x_t \mid c, c_{s_N}, t)$
 4:     $\tilde{x}_t \leftarrow \Psi^t(x_{t-1} \mid c, c_{s_n}, t)$
 5:     $x_{t-1} \leftarrow \Phi^t(\tilde{x}_t \mid c, c_{s_N}, t)$
 6: **end for**
 7: **return** $x_0$

---

## D PROOFS

In this section, we provide a mathematical derivation to validate Eq.8.

**Diffusion Model.** We define $T$ as the total number of denoising steps, $c$ as the conditonal prompt, $c_{s_n}$ as the $n$-th sketch sequence. The denoising process can be defined as $\Phi : \mathcal{N} \times \mathcal{C} \to \mathcal{D}$, where $\mathcal{N}$ is the Gaussion distribution and $\mathcal{D}$ is the target data distribution. Given a starting point $x_T \in \mathcal{N}$, we can generate $x_0 = \Phi(x_T|c, c_{s_n}, \gamma_1) \in \mathcal{D}$, where $\gamma_1$ is the condition guidance scale during denoing. The mapping corresponds $\Phi$ to the probability $P(x_0|c, c_{s_n}, \gamma_1, x_{1:T})$. For simplicity, we only consider the initial input $x_T$ in $\Phi$. Similarly, we can reverse the process using an inversion function $\Psi : \mathcal{D} \times \mathcal{C} \to \mathcal{N}$. Under a inversion guidance scale $\gamma_2$, we obtain inverted data $\tilde{x}_T = \Psi(\tilde{x}_0|c, c_{s_n}, \gamma_2) \in \mathcal{N}$ from $\tilde{x}_0 \in \mathcal{D}$. Following DDPM (Ho et al., 2020), we treat diffusion model as a Monte Carlo process and decompose $\Phi$ into $T$ times single-step denoising mappings as

$$\Phi(x_T|c, c_{s_n}, \gamma_1) = \Phi^T(x_T|c, c_{s_n}, \gamma_1) \circ \cdots \circ \Phi^2(x_2|c, c_{s_n}, \gamma_1) \circ \Phi^1(x_1|c, c_{s_n}, \gamma_1). \tag{9}$$

And we define $\Phi^T$ as

$$x_{t-1} = \Phi^t(x_t|c, c_{s_n}, \gamma_1) = \sqrt{\alpha_{t-1}}\frac{x_t - \sqrt{1-\alpha_t}\epsilon_\theta^t(x_t)}{\sqrt{\alpha_t}} + \sqrt{1-\alpha_{t-1}}\epsilon_\theta^t(x_t), \tag{10}$$

where $\alpha_t := \prod_{i=1}^t (1-\beta_i)$ and $\beta_t$ are the pre-defined parameters for scheduling the scales of adding noises in DDIM (Song et al., 2021a) scheduler. We denote $\epsilon_\theta^t(x_t)$ as the predicted score by the denoising network $\theta$ at timestep $t$.

Similarly, we obtain $\tilde{x}_{t-1}$ as

$$\tilde{x}_t = \Psi^t(\tilde{x}_{t-1}|c, c_{s_n}, \gamma_2) = \sqrt{\frac{\alpha_t}{\alpha_{t-1}}}\tilde{x}_{t-1} + \sqrt{\alpha_t}\left(\sqrt{\frac{1}{\alpha_t}-1} - \sqrt{\frac{1}{\alpha_{t-1}}-1}\right)\epsilon_\theta^t(\tilde{x}_{t-1}). \tag{11}$$

In Eq. 11, we approximate the score predicted at timestep $t$ with timestep $t-1$ along the inversion path, *i.e* set $\epsilon_\theta^t(\tilde{x}_{t-1}) \approx \epsilon_\theta^t(\tilde{x}_t)$. When the approximation error is negligible, $\Phi$ and $\Psi$ can be proven to be inverse functions (Mokady et al., 2023), meaning that $\Psi = \Phi^{-1}$. Therefore, we can represent $\tilde{x}_t$ as

$$\tilde{x}_0 = \Psi^{-1}(\tilde{x}_T|c, c_{s_n}, \gamma_2) = \Phi(\tilde{x}_T|c, c_{s_n}, \gamma_2). \tag{12}$$

**Classifier free guidance.** In the classifier-free guidance approach (Ho & Salimans, 2021), the score prediction model $u_\theta$ is trained rained both conditionally and unconditionally. At inference time, the denoising score is obtained by blending the conditional and unconditional outputs of $u_\theta$ ,allowing flexible control over the strength of guidance through a tunable scale. Specifically, for denoising and inversion process, we use guidance scales $\gamma_1$ and $\gamma_2$, with the corresponding scores as Eq. 7, where $u_\theta$ is the noise predictor, and $\emptyset$ is the null prompt, representing the denoising result under unconditional settings.

**Derivation.** Given inference timestep of $T$, we can obtain the inverted latent $\tilde{x}_T$ as

$$\tilde{x}_T = \sqrt{\frac{\alpha_T}{\alpha_{T-1}}}\tilde{x}_{T-1} + \sqrt{\alpha_T}\left(\sqrt{\frac{1}{\alpha_T}-1} - \sqrt{\frac{1}{\alpha_{T-1}}-1}\right)\epsilon_\theta^T(\tilde{x}_{T-1}). \tag{13}$$

For the sake of convenience, we set

$$m_T = \sqrt{\frac{\alpha_T}{\alpha_{T-1}}}, \quad n_T = \sqrt{\alpha_T}\left(\sqrt{\frac{1}{\alpha_T}-1} - \sqrt{\frac{1}{\alpha_{T-1}}-1}\right). \tag{14}$$

Through iterative and combinatorial processes, $\tilde{x}_T$ could be expressed as

$$\begin{aligned}
\tilde{x}_T &= m_T\tilde{x}_{T-1} + n_T\epsilon_\theta^T(\tilde{x}_{T-1})\\
&= m_T m_{T-1}\tilde{x}_{T-2} + m_T n_{T-1}\epsilon_\theta^{T-1}(\tilde{x}_{T-2}) + n_T\epsilon_\theta^T(\tilde{x}_{T-1})\\
&= m_T m_{T-1} m_{T-2}\tilde{x}_{T-3} + m_T m_{T-1} n_{T-2}\epsilon_\theta^{T-2}(\tilde{x}_{T-3}) + m_T n_{T-1}\epsilon_\theta^{T-1}(\tilde{x}_{T-2}) + n_T\epsilon_\theta^T(\tilde{x}_{T-1})\\
&= \prod_{i=0}^T m_i\tilde{x}_0 + \sum_{t=1}^T n_t \prod_{k=t+1}^T m_k\epsilon_\theta^t(\tilde{x}_{t-1}).
\end{aligned} \tag{18}$$

$$\tag{15}$$

Similarly, we can perform iterative derivations to obtain the equivalent form of $x_T$ as

$$x_T = \prod_{i=0}^{T} m_i x_0 + \sum_{t=1}^{T} n_t \prod_{k=t+1}^{T} m_k \epsilon_\theta^t(x_t). \tag{16}$$

Then in Z-sampling, we focus solely on local cycle of $x_t \to x_{t-1} \to \tilde{x}_t$. Substituting Eq. 9 into Eq. 10 yield $\tilde{x}_t$ as

$$\begin{aligned}
\tilde{x}_t &= x_t - \sqrt{1 - \alpha_t}\epsilon_\theta^t(x_t) + \sqrt{\frac{(1-\alpha_{t-1})\alpha_t}{\alpha_{t-1}}}\epsilon_\theta^t(x_t) \\
&\quad + \sqrt{\alpha_t}\left(\sqrt{\frac{1}{\alpha_t} - 1} - \sqrt{\frac{1}{\alpha_{t-1}} - 1}\right)\epsilon_\theta^t(\tilde{x}_{t-1}) \\
&= x_t + \sqrt{1 - \alpha_t}\left(\epsilon_\theta^t(\tilde{x}_{t-1}) - \epsilon_\theta^t(x_t)\right) + \sqrt{\frac{(1-\alpha_{t-1})\alpha_t}{\alpha_{t-1}}}\left(\epsilon_\theta^t(x_t) - \epsilon_\theta^t(\tilde{x}_{t-1})\right) \tag{17} \\
&= x_t + \left(\sqrt{1 - \alpha_t} - \sqrt{\frac{(1-\alpha_{t-1})\alpha_t}{\alpha_{t-1}}}\right)\left(\epsilon_\theta^t(x_t) - \epsilon_\theta^t(\tilde{x}_{t-1})\right) \\
&= x_t + \sqrt{\alpha_t}\left(\sqrt{\frac{1}{\alpha_t} - 1} - \sqrt{\frac{1}{\alpha_{t-1}} - 1}\right)\left(\epsilon_\theta^t(x_t) - \epsilon_\theta^t(\tilde{x}_{t-1})\right).
\end{aligned}$$

We define the latent difference of Z-Sampling is accumulated as

$$\begin{aligned}
\delta_{Z-\text{Sampling}} &= \sum_{t=1}^{T}(x_t - \tilde{x}_t)^2 \\
&= \sum_{t=1}^{T}\alpha_t h_t^2\left(\epsilon_\theta^t(x_t) - \epsilon_\theta^t(\tilde{x}_{t-1})\right)^2 \tag{18} \\
&= \sum_{t=1}^{T}\alpha_t h_t^2\left(\underbrace{\epsilon_\theta^t(x_t) - \epsilon_\theta^t(\tilde{x}_t)}_{\tau_1:\text{semantic information gain term}} + \underbrace{\epsilon_\theta^t(\tilde{x}_t) - \epsilon_\theta^t(\tilde{x}_{t-1})}_{\tau_2:\text{approximation eror term}}\right)^2.
\end{aligned}$$

Excluding the approximation error introduced by inversion algorithm, we can rewrite Eq. 18 as

$$\delta_{Z-Sampling} = \sum_{t=1}^{T}\alpha_t h_t^2\left(\epsilon_\theta^t(x_t) - \epsilon_\theta^t(\tilde{x}_t)\right)^2. \tag{19}$$

Thus, we have demonstrated that $\delta_{Z-Sampling} \propto \sum_{t=1}^{T}(\tau_1(t))^2$. Although the step-by-step approach results in $x_t$ and $\tilde{x}_t$ being the same at each timestep $t$, from Eq. 7, we note that $\epsilon_\theta^t(x_t)$ and $\epsilon_\theta^t(\tilde{x}_t)$ are obtained under guidance scales $\gamma_1$ and $\gamma_2$ respectively. Thus the effect of Z-sampling is further equivalent as

$$\begin{aligned}
\delta_{Z-Sampling} = \sum_{t=1}^{T}\alpha_t h_t^2\big((1+\gamma_1)u_\theta(x_t, c, c_{s_N}, t) - \gamma_1 u_\theta(x_t, \emptyset, c_{s_N}, t) \\
- (1+\gamma_2)u_\theta(\tilde{x}_t, c, c_{s_n}, t) + \gamma_2 u_\theta(\tilde{x}_t, \emptyset, c_{s_n}, t)\big)^2
\end{aligned} \tag{20}$$

In our experiments, we adopted the parameter settings from ZigZag, setting $\gamma_2 = 0$, which simplifies the above equation as follows:

$$\delta_{Z-Sampling} = \sum_{t=1}^{T}\alpha_t h_t^2\left((1+\gamma_1)u_\theta(x_t, c, c_{s_N}, t) - \gamma_1 u_\theta(x_t, \emptyset, c_{s_N}, t) - u_\theta(\tilde{x}_t, c, c_{s_n}, t)\right)^2 \tag{21}$$

At a given time step $t$, since $s_N$ remains constant, thus, $\delta_{Z-Sampling} \propto \sum_{t=1}^{T}(u_\theta(x_t, c, c_{s_n}, t))^2$. At this point, the proof of Eq. 8 has been completed.

## E  VALIDATING THE MODEL'S TRANSFERABILITY TO EDGE-MAP CONDITIONING

To validate the transferability of our model, we applied our model to edge maps. Specifically, we first convert the corresponding images in the COCO (Lin et al., 2014) dataset into edge maps, then transform them into SVGs to obtain separable contour segments. From these segments, we extract a pseudo "initial sketch" $s_1$ for the SGR mechanism. This process does not rely on real stroke order, nor does it require users to provide the drawing process.

As show in Tab. 3 and Fig. 9, the experimental results show that, the performance in the pseudo-sequence mode remains largely comparable. The generated outputs remain stable in terms of structural consistency and aesthetics. These findings demonstrate that our pre-trained model: (1) can operate effectively even with only the final sketch; (2) maintains stable inference performance without stroke sequence information.

Table 3: Performance comparison demonstrating the model's robustness and adaptability when trained on sketches and applied to edge-map inputs.

| Method | ImageReward | HPS v2 | PickScore | LPIPS-Sketch | CLIP-Score |
|---|---|---|---|---|---|
| ControlNet | 0.023 | 24.85 | 20.02 | **0.43** | 26.01 |
| ControlNet-SPO | 0.77 | 28.05 | 22.16 | 0.48 | 26.23 |
| Ours | **0.96** | **30.53** | **22.71** | 0.46 | **26.63** |

In summary, our method demonstrates excellent adaptability and robustness in practical scenarios, maintaining high-quality generation performance even without real drawing sequences.

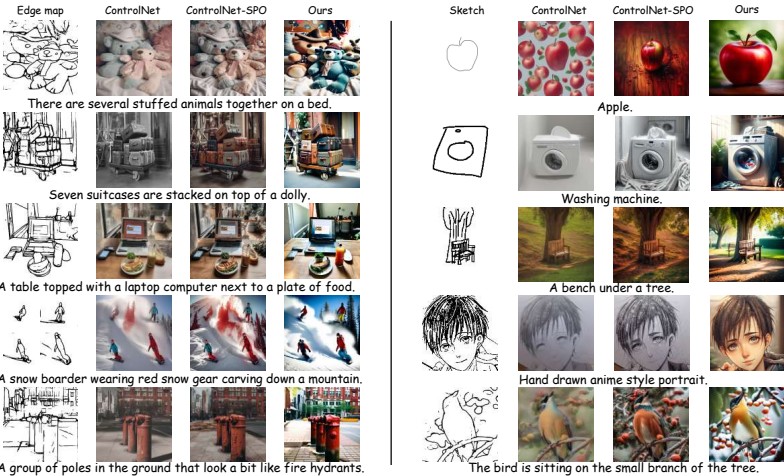

Figure 9: Left: images generated under edge-map guidance. Right: corresponding example images used in the human study.

Table 4: Performance comparison of methods across aesthetic, structural, textual alignment, and realism metrics.

| Method | Aesthetics | Structure Match | Text Alignment | Realism / Naturalness | overall |
|---|---|---|---|---|---|
| ControlNet | 3.60 | 4.68 | 4.71 | 4.08 | 0.8% |
| ControlNet-SPO | 4.38 | 4.28 | 4.79 | 4.22 | 16.53% |
| ours | **4.88** | **4.56** | **4.95** | **4.76** | **82.67%** |

## F  HUMAN STUDY

We conducted human study to evaluate objective quality against real user preferences, comparing Ours with ControlNet and ControlNet-SPO. Twenty five "sketch + text" pairs were used, with each model generating corresponding images presented anonymously in random order. Participants rated Aesthetics, Structure Match, Text Alignment, and Realism on a 5-point Likert scale, and provided an overall preference (A/B choice). Fig. 9 shows example images from the study. Tab. 4 indicate

that our method is consistently preferred across metrics, confirming its superior objective quality and alignment with user preferences.

## G MORE SAMPLE IMAGES GENERATED BY SKETCHEVO

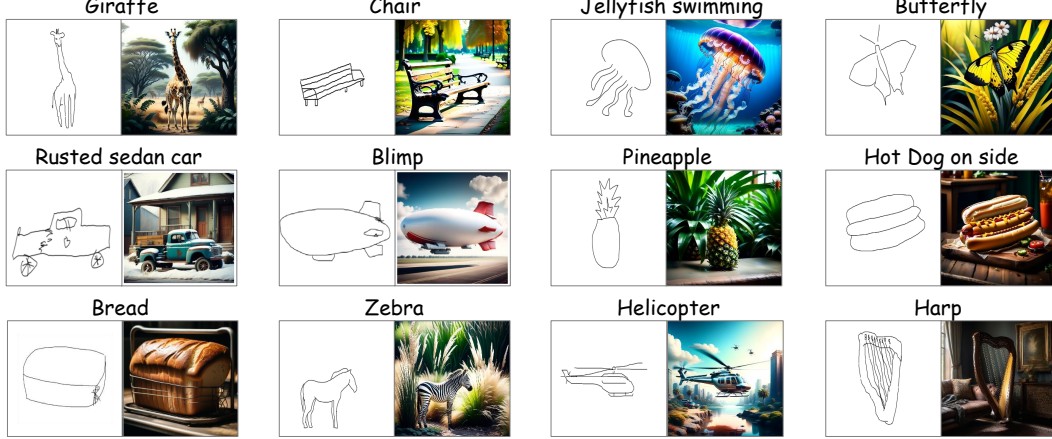

Figure 10: More visualization results for the Sketchy datasets.

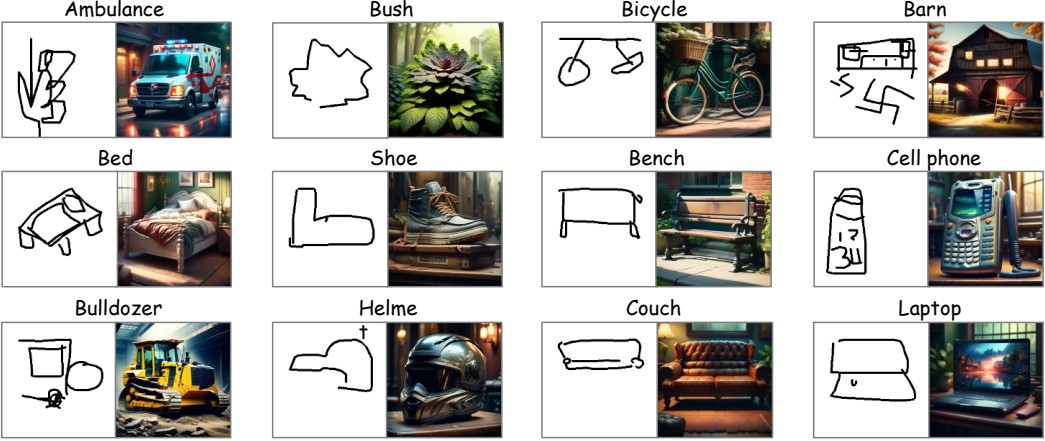

Figure 11: More visualization results for the QuickDraw datasets.

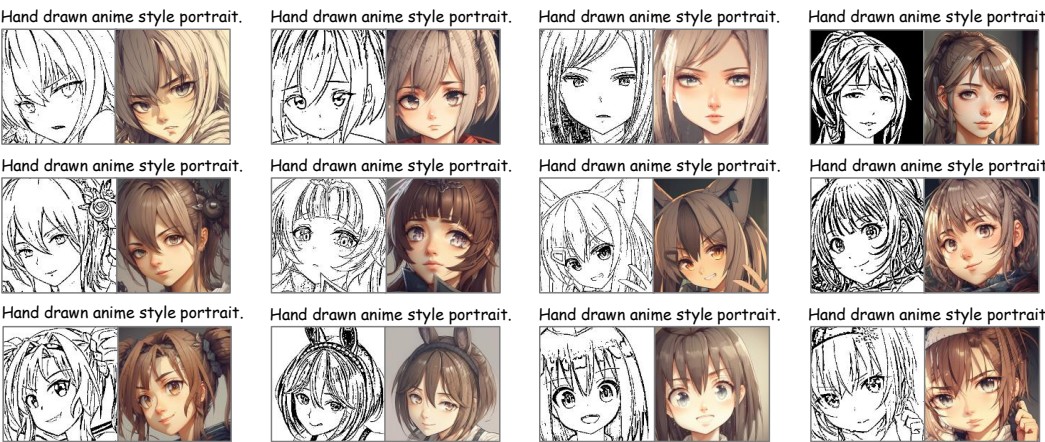

Figure 12: More visualization results for the AnimeDiffusion datasets.

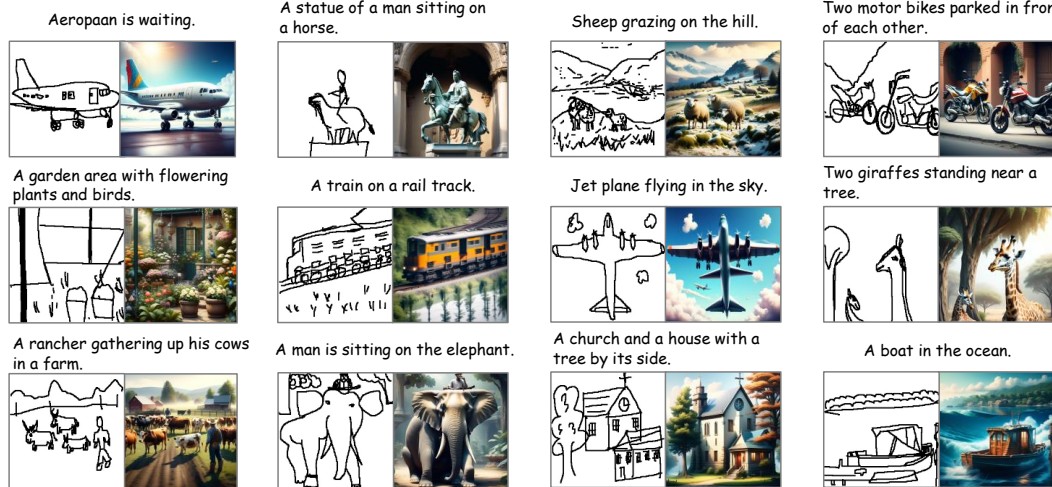

Figure 13: More visualization results for the FSCOCO datasets.

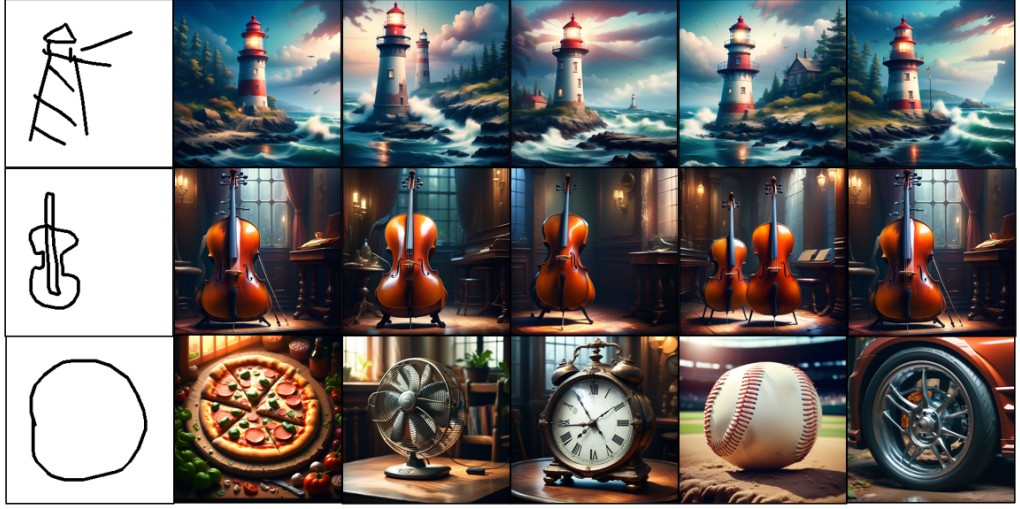

Figure 14: More visualization results off-prompt on the QuickDraw dataset.

## H VISUALIZATION RESULTS OF OFF-PROMPT ON THE QUICKDRAW DATASET

We conduct inference using only the QuickDraw sketches as input. The conclusions are as follows:

**Clear-contour sketches yield consistent structure-driven results.** For sketches with clear and characteristic contours (e.g., lighthouse, cello), the model can still produce semantically aligned images that fit natural scenes even without textual input, suggesting that well-defined structural cues alone can anchor the intended concept.

**Semantic ambiguity in minimalist sketches.** For highly abstract or extremely sparse sketches, removing text introduces substantial semantic uncertainty. For instance, when the sketch is merely a circle, sampling with different random seeds produces diverse and often mutually inconsistent outputs. This demonstrates that textual input is essential for stabilizing and disambiguating the semantics of minimalist drawings.

**Complementary control beyond structure.** In addition to semantics, text typically governs appearance-related attributes such as color, number, material, background composition, lighting, and stylistic preferences—factors not specified by the sketch.

Taken together, these observations highlight the complementary roles of the two modalities: the sketch primarily constrains the structural outline, while the text refines high-level semantics and controls appearance attributes, jointly shaping the final generation.

# I  THE USE OF LARGE LANGUAGE MODELS (LLMS)

In this work, we employed a large language model to assist in polishing the writing and improving the clarity and readability of the text.

