# OpenReview forum: "SketchEvo: Leveraging Drawing Dynamics for Enhanced Image Synthesis"
_ICLR.cc/2026/Conference — ICLR 2026 Poster_

### Official Review · Reviewer_bvYj · 2025-10-22

**Soundness:** 3
**Presentation:** 3
**Contribution:** 3
**Rating:** 6
**Confidence:** 4

**Summary:**

The paper proposes SketchEvo, with two parts:
- SGPO: extend SPO-style pairwise preference optimization to sketch-guided diffusion by building the candidate pool with different sketch stages (not just noise like the prior works) and picking top/bottom by a pretrained preference scorer.

- SGR: an inference-time sequence-guided rollback that blends conditional/unconditional scores with text+sketch (two γ’s), arguing increased “information gain” when text and sketch diverge.

evaluated on Sketchy (train), and tested also on QuickDraw, AnimeDiffusion, FSCOCO. metrics: ImageReward / HPSv2 / PickScore (aesthetics), CLIPScore (text alignment), and a sketch-LPIPS variant (sketch fidelity). baselines include ControlNet, T2I-Adapter, VersaGen, and ControlNet with DPO/SPO.

**Strengths:**

- Simple, intuitive idea: use sequence of sketches as a structured source of diversity ->  stronger positive/negative pairs -> better gradients than noise-only SPO. this feels original within sketch-conditioned alignment.

- Method framing: clear diffusion-probability ratio objective (SPO/DPO family) and a roll-back mechanism adapted to dual conditions (text+sketch).

- Decent empirical lift: consistent gains on preference metrics, with some qualitative improvements in composition/position/color, and cross-dataset demos.

- Ablations: show candidate-pool size effects and performance vs. which sketch stage guides rollback; also a tiny “scoring model” swap test.

- SGPO vs prior alignment: prior DPO/SPO/LPO/D3PO target text-only or step-aware preferences but don’t, to best of my knowledge, explicitly use sketch-sequence stages to create candidate diversity at each diffusion step. that is a novel and plausible extension for multimodal control.

**Weaknesses:**

- SGR vs rollback work: Rollback/zig-zag/self-reflection sampling has been explored for text-to-image; adapting it to joint text+sketch guidance with a sequence-aware blend is a reasonable incremental contribution. the theoretical “information gain increases with text–sketch divergence” is interesting, but depends on approximations and fixed γ’s.

- Baselines: ControlNet, T2I-Adapter, VersaGen are appropriate; ControlNet-DPO/SPO adaptations are helpful. but given 2024–2025 progress, additional strong controls (e.g., ControlNet++ [1] /SmartControl [2] or more recent) would be useful; at least include one recent sketch-aware control baseline if feasible.

- Possible circularity / reward hacking: The same family of preference scorers appears to be used both for pair selection during training and for evaluation, risking overfitting to those scorers rather than to human judgment. Cross-metric generalization is only partially demonstrated.

- Limited human studies / no significance testing: No human pairwise evaluation targeted to sketch-conditioned synthesis. No confidence intervals, multiple seeds, or statistical tests are reported for the main tables.

- Writing clarity: Background part (L126-149), is not clear and detailed enough. All the terms, should be explained clearly for readers to better grasp the discussion. For example, what is $x^w_t$, $x^l_t$ ?

[1] https://arxiv.org/abs/2404.07987

[2] https://arxiv.org/abs/2404.06451

**Questions:**

On reward hacking and evaluation metrics: My primary concern is the potential for "reward hacking" or circularity in the evaluation. The paper states that SGPO uses a "pretrained scoring model" (L195) to select the winning $x^w_t$ and losing $x^l_t$ samples during training. The evaluation in Table 1 then relies on automated preference metrics like ImageReward, HPSv2, and PickScore.

- Could you please clarify which specific "pretrained scoring model" was used for training?
- If this training scorer is one of the evaluation metrics (or from the same family, e.g., another CLIP-based preference model), how can we be sure the model hasn't simply learned to optimize for the biases of this specific scorer, rather than for genuine, generalizable human aesthetic preference?


Necessity of human evaluation: Following the previous point, the gains on automated preference scores are strong, but their correlation to actual human judgment is not guaranteed, especially in a dual-condition (sketch+text) setting.
 - Given the risk of reward hacking, a human evaluation study (e.g., pairwise preference tests comparing "Ours" vs. "ControlNet-SPO" and "ControlNet") seems crucial to validate the paper's central claims. Would it be possible for the authors to provide such a study, even on a smaller scale, to confirm that the observed metric improvements translate to tangible, human-perceived quality gains?


Missing SOTA baseline comparisons: The related work section and introduction cite several strong, recent methods for controllable generation, such as SmartControl  and ControlNet++, which are highly relevant.

- Could the authors provide a quantitative or qualitative comparison against these (or similarly strong) recent baselines? If this is not feasible, could you please provide a more detailed discussion on how SketchEvo's contributions differ from and are expected to perform against these methods, which also aim to improve fidelity in controllable generation?


SGPO Mechanism and Learning Signal: In the SGPO formulation (Eq. 4), the model is trained on pairs ($x^w_t$, $x^l_t$) generated from different sketch conditions ($c_{s_w}$,  $c_{s_l}$)
How does this process ensure the model is learning a general "aesthetic preference" rather than simply learning to "prefer more complete sketches"? For instance, if $x^w_t$ is frequently generated from a late-stage sketch ($s_N$ ) and  $x^l_t$ from an early-stage sketch ($s_1$), is it possible the model is just learning to assign a higher likelihood to more detailed conditions?


Justification for SGR using $s_1$: The ablation (Fig. 5a) suggest that using the most abstract sketch ($s_1$) for the rollback mechanism (SGR) is optimal, attributing this to maximized "information gain" from text-sketch divergence.
- This is an interesting finding, but the theoretical justification in Appendix D relies on several approximations. Could you provide more intuition on why maximal divergence (using $s_1$) is superior to using a more structurally-informative intermediate sketch (e.g., $s_{0.6N}$ )? One might expect a more complete sketch to provide a better structural prior during the rollback.

---

> ### Author Response · Authors · 2025-11-20
>
> We sincerely thank you for your valuable comments. We have revised the relevant descriptions in the paper and re-uploaded the updated version. Our point-by-point responses are below:
>
> ### **About the reward hacking.**
>
> **Clarification on the pretrained scoring model for training.**
>
> We follow the training paradigm of SPO (Step-wise Preference Optimization), using PickScore as the pretrained scoring model. PickScore (v1, publicly available on Hugging Face) is fine-tuned on CLIP-H and trained on hundreds of thousands of real user preference pairs from Pick-a-Pic v1. To handle “noisy intermediate images + sketch conditioning,” we extend PickScore with an additional loss term, enabling it to evaluate image pairs at different diffusion timesteps, rather than only final images.
>
> **The concern of optimizing scorer biases rather than genuine human preferences.**
> We acknowledge that this concern is well-founded, but the risk is constrained by multiple aspects of our design:
> - **Strong alignment with human preferences**: PickScore shows high correlation with human judgments, serving as a reliable proxy for aesthetic preferences.
>
> - **Large-scale, diverse training data**: Pick-a-Pic v1 covers varied scenarios and styles, ensuring generalizable preference learning rather than overfitting specific cases.
>
> - **Cross-metric validation**: In Table 1, we also report HPSv2 and ImageReward, from different model families. Together with human study, these results confirm that our model learns generalizable preferences, not just to maximize PickScore.
>
> ### **About human study .**
> We conducted a small-scale, strictly controlled user study comparing our method with ControlNet and ControlNet-SPO. Using 25 “sketch + text” input pairs, each model generated images that were evaluated by 30 users in a blind test. Images were presented anonymously in random order, and participants rated them on four 5-point Likert-scale dimensions—Aesthetics, Structure Match, Text Alignment, and Realism/Naturalness—while also providing an overall A/B preference. The study covered both single-condition and dual-condition settings. The human study illustration is shown in Fig. 8 (right), and the averaged results are summarized below.
>
> | Method       | Aesthetics | Structure Match | Text Alignment | Realism/Naturalness | Overall Preference |
> |--------------|-----------:|----------------:|---------------:|--------------------:|-------------------:|
> | ControlNet   |        3.6 |            4.68 |           4.71 |                4.08 |              0.80% |
> | ControlNet-SPO |       4.38 |            4.28 |           4.79 |                4.22 |             16.53% |
> | Ours         |       **4.88** |            **4.56** |           **4.95** |                **4.76** |             **82.67%** |
>
> The results demonstrate that participants clearly preferred our method across all dimensions. **This confirms that  improvements in automated metrics translate to tangible human-perceived quality gains. And our generated images better align with human preferences, achieving higher scores in both aesthetics and structural consistency compared to the other models.**
>
> ### **About significance testing.**
> We conducted statistical significance testing across various metrics, and all p-values are well below 0.05 (e.g., ImageReward, HPS v2, PickScore, LPIPS-Sketch, and CLIP-Score are all **below 1E-8**), indicating strong significance. We also report narrow 95% confidence intervals, showing that our results are **stable and reliable**. These findings confirm that performance improvements are statistically significant and robust.
>
> |                | Statistical significance | Confidence interval |
> |----------------|--------------------------|---------------------|
> | ImageReward    | 1.04E-44 ***             | 1.18±0.032          |
> | HPS v2         | 1.3E-8 ***               | 30.08±0.0023        |
> | PickScore      | 5.9E-14 ***              | 22.41±0.038         |
> | LPIPS-Sketch   | 6.7E-76 ***              | 0.15±0.0025         |
> | CLIP-Score     | 1.9E-16 ***              | 24.15±0.073         |
>
> Note: *** indicates p < 0.001 (statistically significant difference)

---

> > ### Author Response · Authors · 2025-11-20
> >
> > ### **About more SOTA result.**
> > We have revised the Related Work section to include a comprehensive discussion of recent sketch-specific approaches, particularly ControlNet++ and SmartControl.
> >
> > Since SmartControl does not provide publicly available trained weights, direct quantitative comparison is not feasible. To ensure a robust evaluation, we expanded our comparisons on the Sketchy dataset to include recent competitive methods — **UniControl, ControlNet++, and AnimateDiff.**
> > **The results demonstrate that our approach maintains strong performance against these baselines.** We have also supplemented the manuscript with detailed qualitative Fig.3 and quantitative analyses Tab.1, providing comprehensive evidence of our method’s capabilities. The comparative results on the Sketchy dataset are presented below:
> >
> > |                | ImageReward | HPS v2 | PickScore | LPIPS-Sketch | CLIP-Score |
> > | :------------: | :---------: | :----: | :-------: | :----------: | :--------: |
> > |  ControlNet++  |   -0.011    | 24.19  |   20.56   |     0.13     |   23.83    |
> > |   AnimateDiff  |    0.23     | 23.68  |   20.42   |     0.14     |   23.56    |
> > |   UniControl   |    0.024    | 23.08  |   20.39   |     0.13     |   23.75    |
> >
> > ### **About SGPO mechanism and learning signal.**
> > SGPO does not equate sketch completeness with preference.
> > **Randomized sketch-stage sampling ensures diversity.** During training, positive and negative samples are drawn from a mixture of early-, mid-, and late-stage sketches, rather than always pairing late-stage sketches as “better.” This prevents the model from learning a trivial “more complete = preferred” signal.
> > **Preference is learned from relative comparisons, not absolute stage.** The loss update uses only the relative preference between paired samples, without reusing previous optimal or worst samples. Continuous resampling and re-comparison across sketch stages enable the model to capture aesthetic features stable across conditions.
> > **Empirical validation confirms generalization.** Our experiments show that the model can generate high-quality results even from weak or sparse sketches, indicating that the learned preference reflects structural consistency and global aesthetic quality, not merely sketch detail.
> >
> > ### **About the role of $ s_1 $.**
> > The **forward generation stage requires the complete sketch** $ s_N $ to ensure structural fidelity, which **aligns with human expectations**. In the SGR rollback stage, **the goal shifts to preference alignment**, requiring maximal information gain. In ZigZag rollback, empty text provides the highest information gain; under sketch conditions, the most abstract sketch $ s_1 $ exhibits the largest divergence from the complete sketch’s structure, producing the strongest supervision signal to guide the model in optimizing aesthetic preferences. Without any sketch condition, rollback degenerates to empty text, which mainly reinforces the text condition and can conflict with improving adherence to the sketch.
> >
> > Appendix D theoretically shows that the SGR information gain $ \delta_{\text{Z-Sampling}}$  is positively correlated with the text-sketch divergence, and using $ s_1 $ maximizes $ \delta_{\text{Z-Sampling}}$ . Ablation experiments confirm that $ s_1 $ achieves the best performance on aesthetic metrics while also improving structural consistency. The forward stage ensures structural fidelity with the complete sketch, while the rollback stage uses $ s_1 $ as a minimal structural anchor to amplify preference signals, achieving both structural fidelity and aesthetic enhancement.
> >
> > If we have misunderstood your question, please kindly clarify, and we will provide a revised response as soon as possible.

---

### Official Review · Reviewer_1prC · 2025-10-28

**Soundness:** 2
**Presentation:** 3
**Contribution:** 3
**Rating:** 4
**Confidence:** 5

**Summary:**

The paper introduces SketchEvo, a sketch-guided image generation framework that leverages the drawing sequence—from early strokes to the completed sketch—rather than treating the final sketch as a static constraint. The core claim is that existing text+sketch methods underperform because they ignore preference signals implicit in the sketching process and struggle to produce sufficiently diverse positive/negative samples for preference alignment under multimodal constraints.

SketchEvo tackles this with two components:

* Sequence-Guided Preference Optimization, which uses sketches at different completion stages to create meaningfully divergent training pairs and yield stronger gradients for preference learning;

* Sequence-Guided Rollback, an inference-time mechanism that balances textual semantics with sketch structure by integrating the sketch sequence into rollback updates so that learned aesthetic preferences transfer to generation while preserving structural fidelity.

**Strengths:**

* Clever use of the sketching sequence (early→late strokes) to create structured candidate diversity for preference learning; pairing SGPO with SGR is a fresh, problem-driven combo.

* Method is well-specified; ablations on candidate-pool diversity and rollback settings support the mechanism(however, they didn't mention the number of GPUs that they used for training and just mentioned the type of GPU); consistent wins on preference and sketch-fidelity metrics across multiple datasets.

**Weaknesses:**

* **Lack of related work & baselines for hand-drawn sketches.** The paper mostly compares to ControlNet-style direct sketch conditioning. It omits stronger, recent sketch-specific approaches such as Democratising Sketch Control [2] and KnobGen [1], which explicitly address amateur/sparse sketches—neither discussed nor used as baselines. Add these to Related Work and include at least one competitive replication.

* **Bad Experiments, need more robust results**: As the author shows in Figure 6, s_i affects the generation. ControlNet and the T2I-Adapter have coefficients that the client can control to receive a different level of condition on spatial information. Did the author change the coefficient for those baselines?

* **Proxy metrics only for aesthetics.** Claims rely on ImageReward/HPSv2/PickScore; no human A/B tests or significance.

* **Requires stroke sequences.** Many users provide only the final sketch. Assess a “no-sequence” mode (pseudo-sequences via edge reveals/stroke segmentation) and report the performance drop.

* **Cost underreported**. SGPO candidate pools and SGR rollbacks add compute. Report wall-clock per step, GPU hours, VRAM, and quality–cost curves vs. K and rollback steps.

* **Effect of text on generation**: I think text has a lot of effect on the generation. We can see this effect in Figure 10. What is the effect of text on generation? Do they have any experiments without text and only rely on the sketch?







[1] Navard, Pouyan, et al. "KnobGen: controlling the sophistication of artwork in sketch-based diffusion models." arXiv preprint arXiv:2410.01595 (2024).

[2] Koley, Subhadeep, et al. "It's All About Your Sketch: Democratising Sketch Control in Diffusion Models." Proceedings of the IEEE/CVF Conference on Computer Vision and Pattern Recognition. 2024.

**Questions:**

* **Metric validity for structure:** Sketch-LPIPS can be biased by edge density. Can you add a complementary metric (e.g., edge chamfer distance, keypoint F-score) and/or a small human “structure-match” study?

* **Training data specifics:** How many unique stroke sequences per class were used? Any filtering for noisy/misaligned strokes? Please clarify sequence quality controls and their impact.

* **Robustness to sketch noise:** How sensitive are results to (a) shuffled stroke order, (b) dropped early/late strokes, (c) jittered stroke geometry, and (d) off-prompt doodles?


**I am open to changing my score based on the author's responses.**

---

> ### Author Response · Authors · 2025-11-20
>
> We are grateful for your detailed comments.  Below is our point-by-point response:
> ### **About more metric validity for structure.**
> To compute the Edge Chamfer Distance (ECD) between sketches and generated images, we first convert the images to grayscale and extract edge points, then calculate the symmetric Chamfer distance between the two sets of points to quantify structural differences. To improve robustness, we experimented with Canny, Sobel, and Laplacian edge detectors, adjusting thresholds, kernel sizes, and applying Gaussian smoothing. Ultimately, we adopted the **Canny + auto threshold + Gaussian preprocessing + symmetric Chamfer distance** approach, which balances contour fidelity and stability.The experimental results are shown below.
>
> | Method   | ControlNet | ControlNet-SPO | Ours  |
> |----------|-----------:|---------------:|------:|
> | ECD      |      15.87 |           9.34 |  7.73 |
>
> Results indicate that our method achieves significantly better adherence to the sketch compared with ControlNet. Although the LPIPS-based structural similarity metric in paper shows slightly lower scores than ControlNet, this does not contradict our findings. Detailed analysis reveals that ControlNet often produces blurrier images, leading to poorly defined edges. This causes the ECD metrics to overestimate the structural error, despite visually less precise adherence to the sketch. In contrast, **our model generates sharper, clearer images, with higher fidelity to the sketch contours and better aesthetic quality.**
>
> Regarding the human "Structure Match" study, see the section on "About human study and statistical significance testing."
> ### **About the details of the training set.**
> In training data construction, **we fix the number of intermediate sketches per category to ensure balanced learning.** Specifically, we uniformly sample $k=5$ sketches at different completion stages, and ablation studies (Fig. 8(a)) show that $k=5$ achieves the best trade-off between training cost and generation quality.
>
> **Minor stroke noise does not affect structural learning.** Since the model takes complete intermediate sketches rather than individual strokes, small local noise has negligible impact on overall structure and learning objectives.
>
> **Preserving natural sketch variations improves robustness.** Human sketches inherently include stylistic differences and stroke irregularities; retaining them helps the model generalize real-world inputs without additional filtering or alignment.
>
> Experimental results confirm that even without filtering, the model reliably captures structural cues and human preference signals.
> ### **About related work and baselines.**
> We have updated the Related Work section to include recent sketch-specific approaches, particularly Democratising Sketch Control [2] and KnobGen [1], which address challenges in amateur and sparse sketches.
>
> Since trained weights for these methods are not publicly available, direct quantitative comparison is infeasible. To ensure robust evaluation, we expanded comparisons on the Sketchy dataset to include **UniControl, ControlNet++, and AnimateDiff.**
>
> **Results show that our approach remains strong against these baselines.**  We have also supplemented the manuscript with detailed qualitative Fig.3 and quantitative analyses Tab.1. The comparative results on the Sketchy dataset are presented below:
>
> |                | ImageReward | HPS v2 | PickScore | LPIPS-Sketch | CLIP-Score |
> | :------------: | :---------: | :----: | :-------: | :----------: | :--------: |
> |  ControlNet++  |   -0.011    | 24.19  |   20.56   |     0.13     |   23.83    |
> |   AnimateDiff  |    0.23     | 23.68  |   20.42   |     0.14     |   23.56    |
> |   UniControl   |    0.024    | 23.08  |   20.39   |     0.13     |   23.75    |
>
> ### **About baseline results.**
> We understand your concern and provide clarification regarding the coefficient settings and the fairness of our experimental design for ControlNet and T2I-Adapter.
>
> 1. **Baseline setup**: We used the official code and pre-trained weights without modifying any coefficients controlling spatial conditioning, ensuring baseline performance reflects their original design.
>
> 2. **Parameter consistency in our method**: Only the SGR module introduces "intermediate sketch selection" and "reverse fallback"; all other parameters, including control configurations, match the baselines, guaranteeing fair comparison.
>
> 3. **Experimental support**: Even without the fallback mechanism and with identical parameters, our model outperforms ControlNet and T2I-Adapter in aesthetic scores, confirming the effectiveness of our core design rather than parameter tuning.

---

> > ### Author Response · Authors · 2025-11-20
> >
> > ### **About the robustness to sketch noise.**
> > In our inference process, the model requires **only a single complete sketch and the first stroke as a minimal anchor,** no additional information about stroke order, intermediate strokes, or minor sketch variations is necessary. As a result, these factors do not significantly affect the generated images, ensuring stable and robust outputs across different sketch conditions.
> >
> > - Shuffled stroke order.
> >
> > During drawing, users often adopt personalized and even non-linear stroke orders. In SketchEvo, **intermediate sketches are useful as structural signals.** In the Sketchy dataset, intermediate sketches come from multiple users with naturally random drawing orders, exposing the model during training to a wide variety of intermediate sketch. This ensures that the model learns to rely on the overall structure and contours encoded in the intermediate sketches rather than on the specific order of strokes. **As a result, even if the stroke order is shuffled at inference, the generated images remain stable and maintain structural and aesthetic consistency.**
> >
> > - Dropped early/late strokes.
> >
> > For dropped early strokes, we can convert the available PNG to SVG to extract $ s_1 $ (the most abstract sketch) for the SGR rollback, ensuring the model still receives effective preference signals and produces stable results.For dropped late strokes, the generation remains unaffected because the model uses an intermediate sketch during inference—any intermediate sketch can serve as the “final” sketch.
> >
> > - Jittered stroke geometry.
> >
> > We sincerely apologize that we are currently unable to provide corresponding experiments or analyses, as no suitable dataset is available and constructing one is challenging. We acknowledge the importance of this scenario and will consider it in future research, aiming to build an appropriate dataset to systematically evaluate and address such cases.
> >
> > - Off-prompt doodles.
> >
> > We have supplemented Appendix F with results generated solely by QuickDraw. The conclusions are as follows:
> >
> > **Clear-contour sketches yield consistent structure-driven results.**
> > For sketches with clear and characteristic contours (e.g., lighthouse, cello), the model can still produce semantically aligned images that fit natural scenes even without textual input, suggesting that well-defined structural cues alone can anchor the intended concept.
> >
> > **Semantic ambiguity in minimalist sketches.**
> > For highly abstract or extremely sparse sketches, removing text introduces substantial semantic uncertainty. For instance, when the sketch is merely a circle, sampling with different random seeds produces diverse and often mutually inconsistent outputs. This demonstrates that textual input is essential for stabilizing and disambiguating the semantics of minimalist drawings.
> >
> > **Complementary control beyond structure.**
> > In addition to semantics, text typically governs appearance-related attributes such as color, number, material, background composition, lighting, and stylistic preferences—factors not specified by the sketch.
> > Taken together, these observations highlight the complementary roles of the two modalities: the sketch primarily constrains the structural outline, while the text refines high-level semantics and controls appearance attributes, jointly shaping the final generation.
> >
> > ### **About human study.**
> > We conducted a small-scale, strictly controlled user study comparing our method with ControlNet and ControlNet-SPO. Using 25 “sketch + text” input pairs, each model generated images that were evaluated by 30 users in a blind test. Images were presented anonymously in random order, and participants rated them on four 5-point Likert-scale dimensions—Aesthetics, Structure Match, Text Alignment, and Realism/Naturalness—while also providing an overall A/B preference. The study covered both single-condition and dual-condition settings. The human study illustration is shown in Fig. 8 (right), and the averaged results are summarized below.
> >
> > | Method       | Aesthetics | Structure Match | Text Alignment | Realism/Naturalness | Overall Preference |
> > |--------------|-----------:|----------------:|---------------:|--------------------:|-------------------:|
> > | ControlNet   |        3.6 |            4.68 |           4.71 |                4.08 |              0.80% |
> > | ControlNet-SPO |       4.38 |            4.28 |           4.79 |                4.22 |             16.53% |
> > | Ours         |       **4.88** |            **4.56** |           **4.95** |                **4.76** |             **82.67%** |
> >
> > he results demonstrate that participants clearly preferred our method across all dimensions. **This confirms that  improvements in automated metrics translate to tangible human-perceived quality gains. And our generated images better align with human preferences, achieving higher scores in both aesthetics and structural consistency compared to the other models.**

---

> ### Author Response · Authors · 2025-11-20
>
> ### **About significance testing.**
> We conducted statistical significance testing across various metrics, and all p-values are well below 0.05 (e.g., ImageReward, HPS v2, PickScore, LPIPS-Sketch, and CLIP-Score are all **below 1E-8**), indicating strong significance. We also report narrow 95% confidence intervals, showing that our results are **stable and reliable**. These findings confirm that performance improvements are statistically significant and robust.
>
> |                | Statistical significance | Confidence interval |
> |----------------|--------------------------|---------------------|
> | ImageReward    | 1.04E-44 ***             | 1.18±0.032          |
> | HPS v2         | 1.3E-8 ***               | 30.08±0.0023        |
> | PickScore      | 5.9E-14 ***              | 22.41±0.038         |
> | LPIPS-Sketch   | 6.7E-76 ***              | 0.15±0.0025         |
> | CLIP-Score     | 1.9E-16 ***              | 24.15±0.073         |
>
> Note: *** indicates p < 0.001 (statistically significant difference)
>
> ### **About the requirements of stroke sequences.**
> To assess the model’s transfering capability, we convert COCO images to edgemaps and then to SVGs to extract separable contour segments, from which we construct a pseudo initial sketch s₁ for the SGR mechanism. This process does not rely on real stroke order or user-provided sequences.
> Experimental results (Fig. 8, left) show that, compared with the real-sequence mode, the pseudo-sequence mode achieves largely comparable performance, with stable structural consistency and aesthetics. **This confirms that our pre-trained model can generate high-quality results even using only the final sketch, demonstrating strong robustness and adaptability in practical scenarios.**
>
> | Method       | ImageReward | HPS v2 | PickScore | LPIPS-Sketch | CLIP-Score |
> |--------------|------------:|-------:|----------:|-------------:|-----------:|
> | ControlNet   |        0.023 |  24.85 |     20.02 |         **0.43** |      26.01 |
> | ControlNet-SPO |       0.77 |  28.05 |     22.16 |         0.48 |      26.23 |
> | Ours         |       **0.96** |  **30.53** |     **22.71** |         0.46 |      **26.63** |
>
> ### **About resource usage**
> We calculated the relevant indicators on the A100:
>
> |                | Wall-clock per step / s | GPU hours / h | VRAM / GB |
> | -------------- | ----------------------- | ------------- | --------- |
> | k = 5          | 1.33                    | 0.019         | 29.75     |
> | Rollback steps | 0.40                    | 0.005         | 9.62      |
>
> ### **References**
> [1] Tversky, Barbara. "What do sketches say about thinking." 2002 AAAI Spring Symposium, Sketch Understanding Workshop, Stanford University, AAAI Technical Report SS-02-08. Vol. 148. 2002.
>
> [2] Eitz, Mathias, James Hays, and Marc Alexa. "How do humans sketch objects?." ACM Transactions on graphics (TOG) 31.4 (2012): 1-10.
>
> [3] Sangkloy, Patsorn, et al. "The sketchy database: learning to retrieve badly drawn bunnies." ACM Transactions on Graphics (TOG) 35.4 (2016): 1-12.

---

### Official Review · Reviewer_tnmo · 2025-10-30

**Soundness:** 3
**Presentation:** 3
**Contribution:** 4
**Rating:** 8
**Confidence:** 3

**Summary:**

SketchEvo seeks to improve sketch-to-image generation by analyzing the drawing process rather than treating sketches as static blueprints. SketchEvo captures the evolutionary stages of drawing—from initial rough strokes to the completed sketch—to understand what the artist prioritizes and intends. The claim is that by recognizing that the sequence and refinement of strokes reveal important information about the drawing's intended meaning, SketchEvo generates images that better align with human preferences and aesthetic sensibilities.

The authors state that the system achieves these improvements through two innovations: a training method that uses sketches at various completion stages rather than random variations to learn aesthetic preferences more effectively, and a generation process that employs initial sketch strokes to guide a "rollback" mechanism balancing structural fidelity with natural beauty.

The results are stated as demonstrating superior aesthetics with higher human preference scores.

**Strengths:**

The notion of capturing artist **intent** in the generation and refinement of an image is a novel idea that this reviewer appreciated. This is a well-written paper. The techniques employed seemed sound. Results were interesting. Ablation studies were good.

**Weaknesses:**

What seemed to be missing is an explicit capturing of the artist's intent with a view towards having the model provide explanations and also learn high level concepts.

**Questions:**

It wasn't clear whether or how the model and the representations captures the artist's intent in an explicit manner. An explanation of that would be helpful.

---

> ### Author Response · Authors · 2025-11-20
>
> We sincerely thank you for bringing attention to this key point concerning the modeling of artist intent in SketchEvo.
> ### **About the explicit explanation of SketchEvo.**
> SketchEvo does not hand-engineer or explicitly annotate “intent features” (e.g., semantic labels or saliency maps). Instead, it leverages the drawing process itself as a high-level representation of intent through two key mechanisms:
>
> **Intermediate sketches as explicit intent signals (SGPO).**
>
> Psychological studies indicate that sketches inherently reflect human cognitive biases and aesthetic preferences. SketchEvo treats intermediate sketches not as low-level features, but as explicit structural signals encoding the artist’s intent during creation. Unlike prior sketch-to-image method that consider sketches as static blueprints, intermediate sketches convey structural intent without any manual specification of “intent.”
>
> **Sequence-Guided Rollback (SGR).**
>
> During inference, SGR uses early sketch stages to guide intermediate denoising, injecting high-level structural intent into the generative trajectory by capturing greater information gain. This explicitly operationalizes intent during generation — a capability not present in existing pipelines.
>
> We recognize that interpretability remains an important direction and will continue to refine our approach to make the model’s understanding of artist intent more transparent and accessible.

---

### Official Review · Reviewer_TQFU · 2025-11-01

**Soundness:** 2
**Presentation:** 3
**Contribution:** 3
**Rating:** 4
**Confidence:** 3

**Summary:**

This paper proposes the SketchEvo framework, which harnesses the dynamic evolution of sketches for enhanced image synthesis. The key insight is that intermediate sketches from different drawing stages represent varying levels of abstraction and detail. These sketches offer meaningful semantic and structural divergence while maintaining a connection to the user's intent.

**Strengths:**

- The paper proposes a novel framework that leverages the sketch sequences for preference-based optimization.
- The experimental results demonstrate improvements over baseline methods.

**Weaknesses:**

- The paper does not clearly justify why using intermediate sketches from different drawing stages enhances human preference for generated images. While intermediate sketches offer more structural divergence and variation, it remains unclear how this divergence and variation improve human aesthetic preference. The early sketch strokes actually contains very little information of the full structure.
- It would be interesting to see the results if the sampling sketch condition $c^k_{s_n}$ were replaced with a constant value $c^k_{s_N}$ while sampling different noise z to ensure sample diversity. To my understanding, this sampling method differs from ControlNet-SPO. This experiment would verify whether the performance gain comes from using intermediate sketches rather than simply increasing the number of samples.

**Questions:**

- Why using intermediate sketches from different drawing stages enhances human preference for generated images?
- Does the performance gain come from using intermediate sketches or simply from increasing the number of samples?

---

> ### Author Response · Authors · 2025-11-20
>
> We sincerely appreciate your relevant and valuable questions. Below are point-by-point responses to the reviewers' key concerns.
> ### **About the role of intermediate sketch.**
> **Intermediate sketches encode human preference cues.** Drawing sketch is a sequence of pen-control actions, and intermediate sketches capture how humans progressively refine structure. Prior work [1–5] shows intermediate sketch  highlight key structural elements and reveal human intent.  Psychological research adds intermediate sketch embodies innate shape/structure preferences, serving as a core vehicle for visual cognition and aesthetic preferences. Thus, intermediate sketches naturally contain preference-related information and are reasonable signals for improving preference alignment.
>
> **Using all sketch stages avoids misleading guidance.**
> The initial sketch, though coarse, is a meaningful stage of drawing. As seen in the camel example (Fig. 3), amateur sketches may accumulate noise, making the final sketch not necessarily more reliable. Relying only on the last sketch can therefore introduce misleading cues. Incorporating sketches from multiple stages provides more stable structural signals and helps the model focus on strokes that truly affect readability and aesthetics.
>
> **Intermediate sketches enhance preference optimization.**
> SPO shows that greater candidate diversity improves preference alignment. As shown in Fig. 5(b), introducing intermediate sketches expands the separation between positive and negative pairs more effectively than noise-only perturbation. Our results confirm this: metrics such as HPS v2 and CLIP-Score improve after including intermediate sketches, indicating that they help the model better capture structural features tied to human preferences.
> ### **About the sources of performance gain.**
> We apologize sincerely for any misunderstanding resulting from our written expression. The performance improvement in our method is indeed driven by intermediate-sketch guidance.
>
> In ControlNet-SPO, only the final sketch  $ c^k_{s_N} $ is used during denoising, and sample diversity comes solely from different Gaussian noises. This is exactly the mechanism you described: fixing the sketch condition while varying only the noise $ z $.  In contrast, our method does not generate more samples by sampling k different noises for the same intermediate sketch.  Instead, we use intermediate sketches from different drawing stages as guidance and pair each with a noise $ z $ to produce the same number $ k $ of samples. The sample count remains unchanged, ensuring a fair comparison.
>
> This verifies that the improvement comes from intermediate sketches, not from having more samples. If we have still misunderstood any part of your question, please let us know—we will address it as soon as possible.
>
> ### **References**
> [1] Ha, David, and Douglas Eck. "A neural representation of sketch drawings." arXiv preprint arXiv:1704.03477 (2017).
>
> [2] Kantrowitz, Andrea. "The man behind the curtain: What cognitive science reveals about drawing." Journal of Aesthetic Education 46.1 (2012): 1-14.
>
> [3] Seibert, Pennie S., and Linda J. Anooshian. "Indirect expression of preference in sketch maps." Environment and behavior 25.4 (1993): 607-624.
>
> [4] Chamberlain, Rebecca, et al. "A dot that went for a walk: People prefer lines drawn with human‐like kinematics." British Journal of Psychology 113.1 (2022): 105-130.
>
> [5] Generation, Idea. "Modeling the Role of Sketching in Design."

---

### Author Response · Authors · 2025-11-26

We sincerely thank all reviewers for their valuable time and constructive feedback. Their insightful suggestions have significantly enhanced the quality of our work.

### **Strength**:
- **Sketch-Guided Preference Optimization**. Our framework leverages intermediate sketches for preference optimization in a simple and novel way, effectively capturing the creator’s intent and enabling new possibilities for multimodal alignment.
- **Joint Design & Mechanism Soundness**. The method is conceptually clear: SGPO enhances aesthetic quality by amplifying positive–negative pair differences, while SGR improves structural consistency by leveraging information gain. Together, they form a coherent framework whose effectiveness is further supported by extensive ablations.
- **Robust Performance & Generalization**. Our results are robust, showing consistent improvements across multiple datasets and outperforming existing approaches in both preference alignment and structural fidelity.
### **Key Clarification**:
- **Effectiveness & Generalization** : We provide additional analyses and human-study results showing that SketchEvo remains stable across diverse sketch types and improves preference alignment and structural coherence.
- **Fairness & Rigor** : We include detailed baseline implementations and settings to ensure fair comparisons and verify the reliability of our results.
- **Assumptions & Applicability** : We show that relying on sketch evolution does not restrict practical usage and that SketchEvo remains robust even under highly abstract or imperfect sketches.
### **Revisions and Additions in the Updated Paper**:
- **Related Work (Sec. 2.1)**: We expanded the discussion to include recent methods such as ControlNet++, SmartControl, and KnobGen, providing clearer context and highlighting the unique contributions of our approach.
- **Problem Statement and Background (Sec. 3)**: We rewrote this section for clarity and added detailed explanations of all variables involved in the formulation.
- **Comparison With State-of-the-Art Methods (Sec. 5.2)**: We incorporated ControlNet++, AnimateDiff, and UniControl as additional baselines, updating the corresponding results in Table 1 and Figure 3.
- **Transferability to Edge-Map Conditioning (Sec. 5.6)**: We added experiments on the COCO dataset to demonstrate the model’s transferability beyond sketch inputs.
- **Human Study (Sec. 5.7)**: We added the full experimental setup and results of the human evaluation to support subjective comparison claims.
- **Visualization on QuickDraw Without Prompts (A.7)**: We added analyses and visualizations showing the model’s behavior under off-prompt conditions using the QuickDraw dataset.

We sincerely appreciate the reviewers’ detailed feedback, which has greatly strengthened our work. We hope that our revisions adequately address the raised concerns and further improve the clarity and robustness of the paper. Thank you again for your time and thoughtful consideration.

---

### Meta-Review · Area_Chair_dRzv · 2026-01-08

**Summary:**

There were skeptical questions around causal attribution of gains, evaluation validity, and baseline completeness, which placed it in "borderline acceptance" initially. However, the authors’ thorough rebuttal about the human study, significance testing, and clarification of SGPO/SGR mechanisms address most of the concerns. The strong corrective evidence convinced the AC, and an "Accept (poster)" is recommended.

**Reviewer Concerns:**

Most of the major concerns, especially those regarding the validity of gains, evaluation rigor, and human preference alignment, were substantially addressed in the rebuttal. The remaining issues, such as Completeness of SOTA baselines, Interpretability, etc., are secondary and would not undermine the core contribution.

**Reviewer Scores:**

The reviewer tnmo is very positive about the paper already with only minor concerns about whether the artists' intent is captured explicitly. This was addressed in the rebuttal. The reviewer bvYj may bump up the rating to 7-8, since the rebuttal explicitly resolved the concerns such as reward hacking, lack of human evaluation, missing significance testing, and baseline strength. Last, the critical concerns from TQFU and 1prC were also addressed (or partially addressed), which would raise the final rating of these skeptical reviewers.

---

### Decision · Program_Chairs · 2026-01-26

Accept (Poster)